# Task-Specific Exploration in Meta-Reinforcement Learning via Task Reconstruction

## Abstract

Reinforcement learning trains policies specialized for a single task. Meta-reinforcement learning (meta-RL) improves upon this by leveraging prior experience to train policies for few-shot adaptation to new tasks. However, existing meta-RL approaches often struggle to explore and learn tasks effectively. We introduce a novel meta-RL algorithm for learning to learn task-specific, sample-efficient exploration policies. We achieve this through task reconstruction, an original method for learning to identify and collect small but informative datasets from tasks. To leverage these datasets, we also propose learning a meta-reward that encourages policies to learn to adapt. Empirical evaluations demonstrate that our algorithm achieves higher returns than existing meta-RL methods. Additionally, we show that even with full task information, adaptation is more challenging than previously assumed. However, policies trained with our meta-reward adapt to new tasks successfully.

## 1 Introduction

Research in reinforcement learning (RL) has led to many impressive achievements and powerful methods in the past decade (Sutton, 2018). However, real-world usage is still not widespread, as the algorithms used to solve RL problems often suffer from low sample efficiency and poor generalization to new tasks (Hospedales et al., 2021; Beck et al., 2023b). Meta-reinforcement learning (meta-RL) is a re-emerging approach that tackles these issues. It follows the meta-learning approach of 'learning to learn' (Schmidhuber, 1987; Thrun & Pratt, 1998), making it well-suited for few-shot adaptation settings. In few-shot adaptation, agents aim to become optimal in any new task from a given distribution of tasks, after collecting only a few episodes. Instead of directly learning to solve tasks from scratch, meta-RL finds RL algorithms that quickly learn the optimal policy. While meta-RL agents are general, the algorithms they produce are task-specific. These meta-learned algorithms contain prior knowledge of the task distribution. By leveraging this prior, they quickly learn the optimal method of exploring or solving a task. Therefore, for any task in the distribution, meta-learned algorithms are expected to be more sample-efficient than and outperform standard RL algorithms.

Meta-RL has already been successfully applied to ad hoc teamwork (Zintgraf et al., 2021a; He et al., 2023; Mirsky et al., 2022), robotics (Nagabandi et al., 2018; Zhao et al., 2022), human-robot interaction (Gao et al., 2019; Ballou et al., 2023), multi-agent RL (Yang et al., 2022; Xu et al., 2022), and sim-to-real transfer (Arndt et al., 2020). Despite this, meta-RL has not yet fully addressed the challenges it aims to overcome (Stoican et al., 2023). While recent years have seen several improvements, meta-RL still struggles with adapting to complex tasks. A significant issue is the difficulty of designing sample-efficient exploration strategies for solving complex, dynamic RL environments. The advantage of meta-RL is that these strategies do not have to be manually crafted, but can be learned from data for each particular task distribution. Often, meta-RL agents interact with a task for a few episodes, before being evaluated on that same task. These agents can be seen as exploring the task by using an implicit or explicit meta-learned exploration strategy. However, implicit exploration may be too sample inefficient for few-shot adaptation (e.g., agents that are not using their task distribution priors effectively during exploration). Recent approaches have shown that explicitly optimizing for sample-efficient exploration leads to more powerful few-shot adaptation agents (Beck et al., 2023b). Moreover, the difficulties of few-shot adaptation extend beyond task exploration. Our empirical

results complement those of Beukman et al. (2024) in showing that optimizing a policy that leverages learned priors to solve new tasks can be non-trivial.

To get a clear high-level picture of how in-context meta-RL agents operate, we consider separating it into three main parts, as shown in Fig. 1. The first phase is task exploration. After a task is sampled from the task distribution, a policy $\pi^{\text{explore}}$ collects a "few" episodes of data from it. Optionally, this policy may be guided by an encoder $f_u$. The goal of $\pi^{\text{explore}}$ is to collect data that contains useful information about the task. This information is then extracted by a second encoder $g$ during the task learning stage, and encoded into a task context vector. Finally, in the task-solving phase, a policy $\pi$ attempts to optimally solve the task at hand. This final stage is similar to standard RL, except that $\pi$ is conditioned on the task context.

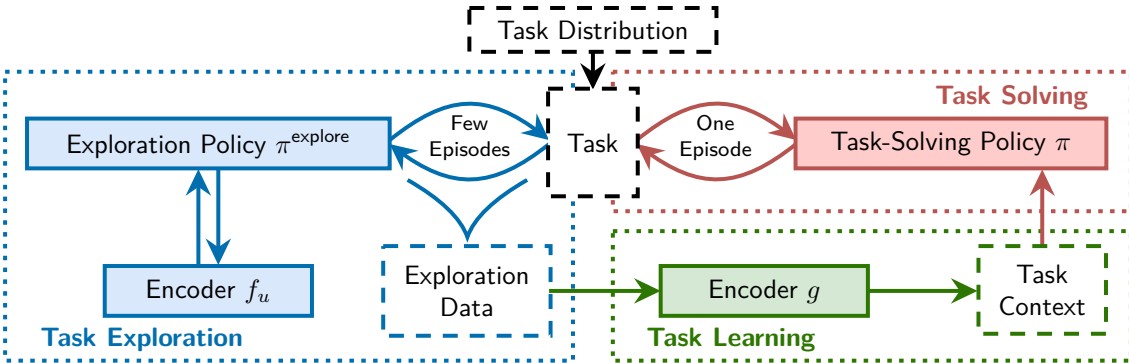

Figure 1: A high-level overview of in-context meta-RL.

There are several ways of learning to reinforcement learn. The approach previously described, and used in this work, is called in-context meta-RL. In-context policies $\pi^{\text{explore}}$ and $\pi$ use task context information provided $f_u$ and $g$ to explore and solve tasks, respectively. Note that this three-phase paradigm is not always cleanly separated in practice, e.g., encoders $f_u$ and $g$ may share information or not be separate encoders at all, a single policy may be used for both task exploration and solving, etc. Nevertheless, this view helps form an intuitive understanding of meta-RL. Besides aiding intuition, we will also be referencing Fig. 1 throughout this paper to clarify which of the three components we are discussing.

We introduce **L**a**t**ent **S**pace **E**xploration via Task **R**econstruction (**LaSER**), a novel meta-RL algorithm for few-shot adaptation.[1] The primary objective of LaSER is to learn task-specific exploration strategies. The core idea behind our method is to identify small datasets that capture rich task-specific information with minimal redundancy. We achieve this by introducing a novel approach of meta-learning by linearly reconstructing these small datasets into much larger datasets collected from the same task. We refer to this process as *task reconstruction*. It serves as a measure of how effective a given small dataset is for few-shot adaptation. Following this measure, LaSER meta-learns a latent space for directly identifying such small datasets, without having to first collect a larger one. This allows training a sample-efficient exploration policy that can collect task-specific data online. We then use the context vectors encoded from this data to train a policy that solves tasks. However, we show that previous approaches to training such a policy fail, even when using optimal task contexts provided by an oracle. Moreover, this failure persists even when the task distribution is replaced by a pre-defined set of tasks. As a result, LaSER's secondary objective is to learn to use task contexts effectively. To achieve this, we propose an augmented RL objective that encourages the task policy to continue exploring until contexts are effectively exploited.

We summarize our main contributions as follows.

1. We formalize an important assumption about the data required for few-shot adaptation. This forms the basis of our task reconstruction method. We then show how an encoder built from these ideas can learn representations that facilitate exploration.

---

[1]Our code is publicly available in an anonymized repository at https://anonymous.4open.science/r/LaSER-anon-4BBB.

2. We leverage these representations to design an intrinsic reward for training a task-specific, sample-efficient exploration policy.

3. We propose using our aforementioned augmented objective to improve the adaptability of task-solving policies through directed exploration of the task context space. This simple change allows meta-RL policies to be trained with standard RL algorithms.

4. We introduce a hybrid encoder architecture composed of a unidirectional and a bidirectional model. The former is used for online task exploration, while the latter offers richer task contexts.

We formalize the meta-RL problem and introduce the necessary background in Sec. 2. In Sec. 3, we address the objectives described above and introduce LaSER, our novel meta-RL algorithm. We position our work within the meta-RL literature in Sec. 4. In Sec. 5, we provide empirical results by evaluating LaSER against existing meta-RL algorithms. We first assess LaSER's performance and adaptability to new tasks after a short exploration phase (i.e., four episodes). We then analyze each individual component, i.e., exploration policy, encoder, and task policy. Finally, we conclude and provide directions for future research in Sec. 6.

## 2   Preliminaries

**Meta-Reinforcement Learning.** Meta-RL extends the standard RL problem. Instead of learning from a single task, a meta-RL agent trains on a distribution of tasks. Usually, the agent's objective is adaptation to new tasks. We consider a probability distribution $p(\mathcal{M})$ over tasks and define each task $\mathcal{M}_i \sim p(\mathcal{M})$ as a Markov decision process (MDP) $(\mathcal{S}, \mathcal{A}, T_i, R_i, \gamma, H)$. All MDPs share the set of states $\mathcal{S}$, set of actions $\mathcal{A}$, discount factor $\gamma$, and horizon $H$. However, the transition function, given as the probability $T_i(s' \mid s, a, \mathcal{M}_i)$ of transitioning from state $s$ to state $s'$ by taking action $a$ in task $\mathcal{M}_i$, is task-dependent. Similarly, the reward function $R_i$ is task-dependent. An episode $\tau = \{s_t, a_t, r_t\}_{t=1}^H$ is a sequence of states, actions, and rewards, such that at each time-step $t$ a tuple $(s_t, a_t, r_t)$ is collected. For a distribution over episodes $P_{\mathcal{M}_i}^\pi(\tau)$, we denote $\tau \sim P_{\mathcal{M}_i}^\pi$ to be an episode collected by following a policy $\pi(a \mid s)$ in task $\mathcal{M}_i$. The return of an episode $\tau$ is the accumulated discounted reward $G(\tau) = \sum_{t=1}^H \gamma^t r_t$.

**In-Context Meta-RL** A common approach to solving meta-RL problems is in-context meta-RL, where a pretrained policy acts as an adaptable RL algorithm (Duan et al., 2016; Laskin et al., 2023; Moeini et al., 2025). Formally, given a context vector $c_i$ that describes a task $\mathcal{M}_i$, the goal is to train a policy $\pi$ to optimally solve $\mathcal{M}_i$ by taking in-context actions $a \sim \pi(\cdot \mid s, c_i)$, conditioned on both the state $s$ and the context $c_i$. The meta-RL agent can therefore learn new policies from $c_i$, even post-training. A standard approach is to concatenate $s$ and $c_i$, then optimize the in-context policy $\pi(a \mid s, c_i)$ using standard RL algorithms. However, this simple method may not generalize well across tasks in complex task distributions (Beukman et al., 2024). To address this, we propose a novel meta-RL optimization algorithm that explicitly uses $c_i$ to train in-context policies.

**Meta-RL for Few-Shot Adaptation.** In few-shot adaptation settings, in-context meta-RL attempts to optimize $\pi$ to solve previously unseen tasks $\mathcal{M}_i \sim p(\mathcal{M})$ in a sample-efficient manner. However, because the meta-RL agent lacks knowledge of the task-specific dynamics $T_i$ and $R_i$, it must first collect data from $\mathcal{M}_i$. The few-shot constraint limits this to $K$ episodes per task. We call this a $K$-shot adaptation problem. Following Beck et al. (2023b), we refer to a sequence of $K$ episodes collected in $\mathcal{M}_i$ as a meta-episode $\mathcal{D}_i^{(K)} = \{\tau_k\}_{k=1}^K$. We can then compute a latent context vector $c_i = g(\mathcal{D}_i^{(K)})$ using an encoder $g$. Importantly, $c_i$ should capture information about $T_i$ and $R_i$, to allow generalization to new tasks. To collect such meta-episodes, we follow Liu et al. (2021) and define a separate exploration policy $\pi^{\mathrm{explore}}$, decoupled from the task-solving policy $\pi$. For a given task, its goal is to collect an informative meta-episode, without necessarily achieving a high return. We define $\pi^{\mathrm{explore}}(a \mid s, \Gamma_i)$ to be an in-context policy, where $\Gamma_i$ is a task representation encoded specifically for exploration. Although we can choose $\Gamma_i$ to be the same as $c_i$, it is less restrictive to use a new representation. Given this setup, we generalize Liu et al. (2021) and consider

agents that maximize the meta-RL objective

$$\mathcal{J}(\pi, \pi^{\text{explore}}, g) = \mathbb{E}_{\mathcal{M}_i \sim p(\mathcal{M}), \mathcal{D}_i^{(K)} \sim P_{\mathcal{M}_i}^{\pi^{\text{explore}}}} \left[ V_i^{\pi} \Big( g(\mathcal{D}_i^{(K)}) \Big) \right], \tag{1}$$

$$\text{where } V_i^{\pi}(\boldsymbol{c}_i) = \mathbb{E}_{\tau \sim P_{\mathcal{M}_i}^{\pi(\boldsymbol{c}_i)}} [G(\tau)]$$

is the expected return in task $\mathcal{M}_i$ for the in-context policy $\pi$ conditioned on $\boldsymbol{c}_i$.

**Training Setup.** Meta-training is the process of training a meta-RL agent. The goal is to learn a sample-efficient algorithm for optimizing policies for each task encountered from $p(\mathcal{M})$. The agent's ability to generalize is evaluated during meta-testing, by solving new tasks from $p(\mathcal{M})$. For each meta-testing task, the agent is first allowed to collect and learn from $K$ episodes. Then, it is evaluated according to Eq. 1. We refer to collecting $K$ episodes from $P_{\mathcal{M}_i}^{\pi^{\text{explore}}}$ as task exploration. This type of exploration is meta-learned to be task-specific. Besides task exploration, the agent still performs the standard RL exploration. To achieve their objectives, $\pi$ and $\pi^{\text{explore}}$ have to explore their environment during meta-training. This is commonly referred to as meta-exploration (Beck et al., 2023b). During meta-testing, there is no meta-exploration.

**Data Representation.** States, actions, and rewards can be represented as vectors or scalars. So, a timestep $(s_t, a_t, r_t)$ is a $d$-dimensional vector, where $d$ is the sum of the dimensions of $s_t$, $a_t$, and $r_t$. We extend this to episodes. For $H' = Hd$, an episode $\tau \in \mathbb{R}^{H'}$ is a vector of $H$ concatenated timesteps. We can further extend this to meta-episodes. For a meta-episode $\mathcal{D}^{(K)} \in \mathbb{R}^{H' \times K}$ composed of $K$ episodes, the $k$-th column $\mathcal{D}_{:,k}^{(K)}$ corresponds to the $k$-th episode. Episodes and meta-episodes can therefore represented as tensors instead of sequences, when convenient.

## 3 Methods

In this section, we present our approach to in-context meta-RL and use the ideas we introduce to design the main components of our proposed algorithm, LaSER. We structure this section based on the three-stage meta-RL paradigm in Fig. 1. Sec. 3.1 corresponds to the task-solving phase, while Secs. 3.2 and 3.3 focus on the task exploration part. Sec. 3.4 continues task exploration but also discusses task learning, with Sec. 3.5 corresponding to the combination of these three stages into a complete meta-RL algorithm.

Specifically, in Sec. 3.1, we introduce a novel approach for in-context RL optimization. Given a context vector $\boldsymbol{c}_i$ for task $\mathcal{M}_i$, we propose augmenting the standard RL objective with a term that encourages meta-exploration whenever $\pi$ can achieve better returns by improving its understanding of $\boldsymbol{c}_i$. Then, Sec. 3.2 discusses the training of an exploration policy $\pi^{\text{explore}}$ for collecting the data required to compute $\boldsymbol{c}_i$. It assumes a given encoder $f_u$ for training $\pi^{\text{explore}}$, with $f_u$ modeled as a unidirectional transformer (Vaswani et al., 2017) of size $d_{\text{model}}$. We propose a novel method of training $f_u$ for sample-efficient task-specific exploration in Sec. 3.3. We first introduce an important assumption on the data collected for few-shot adaptation, then show how a practical learning objective, i.e., task reconstruction, can be built from it. Next, we describe our proposed architecture for encoders $f_u$ and $g$ in Sec. 3.4. We model $g$ as a transformer, as their ability to process each timestep in the context of an entire meta-episode has been shown to be advantageous in meta-RL settings (Melo, 2022; Shala et al., 2024). Finally, in Sec. 3.5, we introduce our meta-training and meta-testing algorithms. From this point forward, we simplify notation and drop the task subscript $i$ whenever it is clear that we are referring to task-specific data or representations.

### 3.1 Task Solving

As discussed in Sec.2, Beukman et al. (2024) show that optimizing an in-context policy $\pi(a_t \mid s_t, \boldsymbol{c})$ to adapt to and solve tasks is non-trivial. We therefore introduce an approach for augmenting rewards with a term that encapsulates performance in the meta-RL setting. The agent can leverage this term to perform directed meta-exploration in the context of $\boldsymbol{c}$. With this new reward, we could overcome the aforementioned limitation by simply optimizing $\pi(a_t \mid s_t, \boldsymbol{c})$ using standard RL algorithms.

We start from the hypothesis that the task policy $\pi$ is suboptimal because it fails to understand that $\boldsymbol{c}$ offers important information about the current task $\mathcal{M}_i$. In a standard RL setting, a policy $\pi(a_t \mid s_t)$ would need

to explore enough to understand the (single) task it is expected to solve. In our meta-RL setting, we have an additional meta-exploration objective. The policy $\pi(a_t \mid s_t, \boldsymbol{c})$ must understand the relationship between $s_t$ and $\boldsymbol{c}$, at any timestep $t$ where the optimal action depends on both $s_t$ and $\boldsymbol{c}$. In other words, $\pi$ needs to meta-explore more than in a standard RL setting.

Following this idea, we propose a method that encourages $\pi$ to meta-explore more if the task context $\boldsymbol{c}$ is not properly used. Let $V(s_t, \boldsymbol{c})$ and $V(s_t)$ be the approximated value functions at state $s_t$, with and without considering $\boldsymbol{c}$, respectively. We make the assumption that, for an optimal policy, $V(s_t) \leq V(s_t, \boldsymbol{c})$ should hold at any timestep $t$. That is, using $\boldsymbol{c}$ should never decrease performance. If this does not hold for $\pi$ in some state $s_t$, then the agent should meta-explore more from $s_t$. We adopt the standard approach of encouraging exploration by maximizing entropy (Williams & Peng, 1991), with the important distinction that this maximization is guided by the context $\boldsymbol{c}$. This idea is implemented as a meta-reward

$$r_t^+ = r_t + \beta\, w(s_t, \boldsymbol{c})\, S[\pi](s_t, \boldsymbol{c}), \tag{2}$$

$$\text{where } w(s_t, \boldsymbol{c}) = \max\left(0, \tanh\left(V(s_t) - V(s_t, \boldsymbol{c}) - \zeta\right)\right) \tag{3}$$

is a non-negative dynamic weight on the entropy $S$ of $\pi(\cdot \mid s_t, \boldsymbol{c})$, $r_t$ is the environment reward, and $\beta$ is a constant. Note that $w(s_t, \boldsymbol{c})$ is non-zero only when $V(s_t) - V(s_t, \boldsymbol{c}) \geq \zeta$, for a given threshold $\zeta \in (-\infty, 0]$.

Intuitively, $w(s_t, \boldsymbol{c}) > 0$ means that the agent is disregarding the information provided by $\boldsymbol{c}$ and should meta-explore more from state $s_t$. When $w(s_t, \boldsymbol{c}) = 0$, i.e., $V(s_t) \leq V(s_t, \boldsymbol{c}) + \zeta$, we recover the original RL objective, with $r_t^+ = r_t$. By simply replacing $r_t$ with $r_t^+$, $\pi$ can now be optimized using standard RL algorithms. We use the proximal policy optimization (Schulman et al., 2017, PPO) algorithm in our work. Furthermore, we stabilize PPO (Wang et al., 2020; Sun et al., 2022; 2023; Moalla et al., 2024) using proximal feature optimization (Moalla et al., 2024, PFO). In Appendix A, we provide a short overview of PPO and explain our method of applying it to meta-RL.

To get a better understanding of the meta-reward in Eq. 2, we analyze its effect on the gradient of the RL objective. Consider a simple policy gradient objective $\mathcal{J}_t(\pi) = \mathbb{E}_{s_t \sim d_t^\pi, a_t \sim \pi_t}[G_t]$ computed at timestep $t$, where $d_t^\pi(s_t)$ is the distribution over states when following policy $\pi$, and $G_t = \sum_{l=0}^{\infty} \gamma^l r_{t+l}$ is the discounted return from action $a_t$ onward. By replacing $r_t$ with $r_t^+$, we can derive the augmented return $G_t^+(\pi) = G_t + \beta \sum_{l=0}^{\infty} \gamma^l w(s_{t+l}, \boldsymbol{c}) S[\pi](s_{t+l}, \boldsymbol{c})$. Given the new meta-RL objective $\mathcal{J}_t^+(\pi) = \mathbb{E}_{s_t \sim d_t^\pi, a_t \sim \pi_t}[G_t^+(\pi)]$, the gradient w.r.t. (the parameters of) $\pi$ is

$$\begin{aligned}
\nabla_\pi \mathcal{J}_t^+(\pi) = \mathbb{E}_{s_t \sim d_t^\pi, a_t \sim \pi_t}\Big[ & G_t \nabla_\pi \log \pi(a_t \mid s_t, \boldsymbol{c}) \\
& + \beta \left(\sum_{l=0}^{\infty} \gamma^l w(s_{t+l}, \boldsymbol{c}) S[\pi](s_{t+l}, \boldsymbol{c})\right) \nabla_\pi \log \pi(a_t \mid s_t, \boldsymbol{c}) \\
& + \beta \sum_{l=0}^{\infty} \gamma^l w(s_{t+l}, \boldsymbol{c}) \nabla_\pi S[\pi](s_{t+l}, \boldsymbol{c}) \Big]
\end{aligned} \tag{4}$$

The first term is the gradient of the standard RL objective $\mathcal{J}_t(\pi)$. The second term weighs the policy gradient by the entropy of $\pi$ in all future visited states, where $w$ acts as a soft on-off switch, ignoring states that have already been meta-explored properly. Intuitively, the update increases the likelihood of taking action $a_t$ if it leads to states in which $\pi$ has both high entropy and a poor understanding of $\boldsymbol{c}$. The agent may therefore be directed towards states that benefit from additional meta-exploration. This is an advantage, as reaching such states more often implies better learning of how $\boldsymbol{c}$ can be used there. The entropy in all of these states is then increased in the third term, which is appropriately weighted by $w$ and $\gamma^l$. This ensures sustained meta-exploration from any $s_{t+l}$ with $w(s_{t+l}, \boldsymbol{c}) > 0$.

This section introduced a novel meta-exploration method for meta-training task policies. However, it does not address task exploration. We tackle this distinct challenge in the next subsection.

### 3.2 Task Exploration Policy

To create a context $\boldsymbol{c}$ for identifying a task, the meta-RL agent must first learn to explore tasks. The objective of the exploration policy $\pi^{\mathrm{explore}}$ is to collect a single informative meta-episode $\mathcal{D}^{(K)}$ for a task $\mathcal{M}_i \sim p(\mathcal{M})$. Specifically, $\pi^{\mathrm{explore}}$ should collect information about the task-specific dynamics of $\mathcal{M}_i$ while avoiding irrelevant or redundant exploration. Ultimately, $\mathcal{D}^{(K)}$ should enable a task policy $\pi$ to become optimal in $\mathcal{M}_i$, assuming that $\pi$ has enough prior knowledge about the task distribution $p(\mathcal{M})$.

The encoder $f_u$ computes a matrix $\boldsymbol{\Gamma} = f_u(\mathcal{D}^{(K)}) \in \mathbb{R}^{(Hd_{\mathrm{model}}) \times K}$. Here, $\boldsymbol{\Gamma}$ is a latent representation of the $K$ episodes in $\mathcal{D}^{(K)}$, with $\boldsymbol{\Gamma}_{:,k}$ denoting the representation of the $k$-th episode $\mathcal{D}^{(K)}_{:,k}$. We optimize $f_u$ such that the similarity between $\boldsymbol{\Gamma}_{:,k}$ and $\boldsymbol{\Gamma}_{:,k'}$ is inversely proportional to how useful it is to collect episode $\mathcal{D}^{(K)}_{:,k'}$, after having already collected episode $\mathcal{D}^{(K)}_{:,k}$, for any $k, k' \in [K], k < k'$.[2] We measure similarity using the pairwise cosine similarity matrix $S_C(\boldsymbol{\Gamma}^{\mathsf{T}}, \boldsymbol{\Gamma}) \in \mathbb{R}^{K \times K}$. Lower scores correspond to lower similarity between the $K$ episodes, which in turn corresponds to $\mathcal{D}^{(K)}$ being more informative. In Sec. 3.3, we show how to meta-train a function $f_u$ with this property.

We use this idea to optimize $\pi^{\mathrm{explore}}$ for task exploration. We begin by computing the average column-wise similarity in $\mathcal{D}^{(K)}$ as a vector $\boldsymbol{d} = \frac{1}{K} S_C(\boldsymbol{\Gamma}^{\mathsf{T}}, \boldsymbol{\Gamma})^{\mathsf{T}} \mathbf{1} \in \mathbb{R}^K$. Here, $\boldsymbol{d}_k$ represents the average similarity between the $k$-th episode and all other episodes. Therefore, a meta-episode $\mathcal{D}^{(K)}$ contains an informative and diverse set of episodes if $\boldsymbol{d} \approx 0$. Based on this, we define an intrinsic reward function for encouraging task exploration. For the $k$-th exploration episode, the agent receives a task-dependent reward

$$\tilde{R}_k(s^k_t, a^k_t) = \begin{cases} \exp\left(-\frac{1}{\sigma} \boldsymbol{d}_k^2\right) & \text{if the } k\text{-th episode terminates at timestep } t, \\ 0 & \text{otherwise,} \end{cases} \tag{5}$$

where $\sigma$ is a constant, and $s^k_t$ and $a^k_t$ denote the state and action, respectively, at timestep $t$ of episode $k$. We use PPO to train $\pi^{\mathrm{explore}}$ to maximize this intrinsic reward. In practice, $\pi^{\mathrm{explore}}$ is meta-trained on full meta-episodes. Therefore, we apply a causal mask when computing $\boldsymbol{d}$ to hide similarities between the representation of an episode $\mathcal{D}^{(K)}_{:,k}$ and those of any future episodes $\mathcal{D}^{(K)}_{:,k'}$, for $k < k'$.

For $\pi^{\mathrm{explore}}$ to collect a useful episode, it must have knowledge of the episodes it has already collected. To create an exploration-focused task context, we denote $\mathcal{D}^{(:k,:t)}$ to be an incomplete meta-episode, containing $k-1$ complete episodes, and the first $t$ timesteps of the $k$-th episode, for $k \in [K], t \in [H]$. We then condition the in-context policy $\pi^{\mathrm{explore}}$ on the history $\boldsymbol{\Gamma}^{(:k,:t)} = f_u(\mathcal{D}^{(:k,:t)})$ and explore episode $k$ by taking action $a^k_{t+1} \sim \pi^{\mathrm{explore}}(\cdot \mid s^k_{t+1}, \boldsymbol{\Gamma}^{(:k,:t)})$. The overall task exploration process is visualized in Fig. 2.

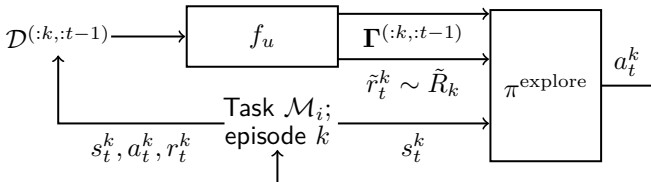

Figure 2: Task exploration. Policy $\pi^{\mathrm{explore}}$ explores a task $\mathcal{M}_i$ by collecting $K$ episodes. At each timestep $t$ of each episode $k$, $\pi^{\mathrm{explore}}$ is conditioned on the current state $s^k_t$ and the latent representation $\boldsymbol{\Gamma}^{(:k,:t-1)}$ of previous timesteps and episodes. Its objective is to take actions that maximize the intrinsic exploration reward function $\hat{R}_k$. Both $\boldsymbol{\Gamma}^{(:k,:t-1)}$ and $\hat{R}_k$ are computed by the encoder $f_u$ from the data collected.

### 3.3 Learning to Explore by Reconstructing Tasks

The final components of our meta-RL algorithms are encoders $g$ and $f_u$, which enable task-solving and task exploration, respectively. Let $\mathbf{B} \in \mathbb{R}^{Q \times H' \times K}$ be the tensor of $Q$ meta-episodes collected from $\mathcal{M}_i$. That is,

---

[2]With a slight abuse of notation, we use $[N]$ to mean the sequence $[1, 2, \dots, N]$ for any integer $N$.

for $j \in [Q]$, $\mathbf{B}_{j,:,:}$ is the $j$-th meta-episode in $\mathbf{B}$. For simplicity, we use $\mathbf{B}_j$ to denote $\mathbf{B}_{j,:,:}$. Additionally, to emphasize the relationship between $\mathbf{B}_j$ and $\mathbf{B}$, we refer to meta-episodes as $\mathbf{B}_j$ instead of $\mathcal{D}^{(K)}$.

Recall that, given $\mathbf{B}_j$, we use $\boldsymbol{c} = g(\mathbf{B}_j)$ for task-solving and $\boldsymbol{\Gamma} = f_u(\mathbf{B}_j)$ for task exploration. The LaSER encoders $g$ and $f_u$ are optimized based on a key assumption about the data used for few-shot adaptation. Before describing this assumption, we first formalize the idea of $K$-shot adaptation.

**Definition 1** ($K$-Shot Adaptation). A policy $\pi$ is $K$-shot adaptable in a distribution $p(\mathcal{M})$ over MDPs if, for any $\mathcal{M}_i \sim p(\mathcal{M})$, $\pi$ becomes optimal in $\mathcal{M}_i$ after at most $K$ episodes collected from $\mathcal{M}_i$.

Finding these $K$ episodes can be difficult. Instead, consider rearranging the tensor $\mathbf{B} \in \mathbb{R}^{Q \times H' \times K}$ into a matrix $\boldsymbol{B}_{[2]} \in \mathbb{R}^{H' \times (QK)}$ via mode-2 matricization (Vasilescu, 2009). That is, $(\boldsymbol{B}_{[2]})_{:,k}$ represents the $k$-th episode, for $k \in [QK]$. If a policy is $K$-shot adaptable in a task $\mathcal{M}_i$ when using the $QK$ episodes in $\boldsymbol{B}_{[2]}$, Definition 1 implies that $\boldsymbol{B}_{[2]}$ contains the $K$ episodes required for $K$-shot adaptation. More precisely, there exists a meta-episode $\boldsymbol{B}_{[2]}^*$ composed of these $K$ episodes. To create a practical algorithm from this idea, we make the following assumption on the relationship between $\boldsymbol{B}_{[2]}$ and $\boldsymbol{B}_{[2]}^*$.

**Assumption 1** (Linear Task Reconstruction). *Let $\boldsymbol{B}_{[2]} \in \mathbb{R}^{H' \times (QK)}$ contain $QK$ episodes collected from an MDP $\mathcal{M}_i$ such that $\boldsymbol{B}_{[2]}$ is sufficient for $K$-shot adaptation. Let $\boldsymbol{B}_{[2]}^* \in \mathbb{R}^{H' \times K}$ be the submatrix of $\boldsymbol{B}_{[2]}$ that contains these $K$ necessary episodes. Then, we assume there exists $\boldsymbol{C}_{[2]} \in \mathbb{R}^{K \times (QK)}$ such that*

$$\boldsymbol{B}_{[2]}^* \boldsymbol{C}_{[2]} \approx \boldsymbol{B}_{[2]}. \tag{6}$$

It follows from Assumption 1 that $\operatorname{rank}(\boldsymbol{B}_{[2]}) \approx K$. Let $\boldsymbol{B}_{[2]}^*$ be the submatrix containing $K$ linearly independent columns of $\boldsymbol{B}_{[2]}$. Then, $\boldsymbol{B}_{[2]}^*$ is an optimal full-rank approximation of $\boldsymbol{B}_{[2]}$, and there exists a coefficients matrix $\boldsymbol{C}_{[2]}$ such that Eq. 6 holds. For efficiency and simplicity, we switch back to tensor representations. We use $\mathbf{B}$ instead of $\boldsymbol{B}_{[2]}$, represent $\boldsymbol{B}_{[2]}^*$ as the meta-episode $\mathbf{B}_j \in \mathbb{R}^{H' \times K}$ for a given $j \in [Q]$, and rearrange $\boldsymbol{C}_{[2]}$ into $\mathbf{C} \in \mathbb{R}^{Q \times K \times K}$. Eq. 6 is then reframed as the batch matrix multiplication $\mathbf{B}_{j,:,:} \mathbf{C}_{l,:,:} \approx \mathbf{B}_{l,:,:}$ for all $l \in [Q]$, which we denote compactly as $\mathbf{B}_j \mathbf{C} \approx \mathbf{B}$.

To identify meta-episodes of interest during task exploration, we introduce a novel approach of reconstructing tasks, based on Assumption 1. For a meta-episode $\mathbf{B}_j$ collected by $\pi^{\text{explore}}$, our method allows us to quantify its effectiveness for $K$-shot adaptation. That is, for a given $j$, we measure how effective it is to compute a task context vector $\boldsymbol{c}$ from $\mathbf{B}_j$ instead of $\mathbf{B}$. Following Eq. 6, the problem is reduced to probing the linear relationship between $\mathbf{B}_j$ and $\mathbf{B}$, by attempting to find coefficients $\mathbf{C}$. To this end, we define the task-reconstruction loss

$$\mathcal{L}_{\text{t-rec}} = \mathbb{E}_{\mathcal{M}_i \sim p(\mathcal{M}),\, \mathbf{B} \sim P_{\mathcal{M}_i}^{\pi^{\text{explore}}},\, j \sim U(Q)} \left[ \operatorname{MSE}\left(\mathbf{B}_j \mathbf{C}, \mathbf{B}\right) \right], \tag{7}$$

where $U(Q)$ is a uniform distribution over the integers $\{1, 2, \ldots, Q\}$, MSE is the mean squared error, and $\mathbf{C}$ is computed from $\mathbf{B}$ and $\mathbf{B}_j$ using encoder $g$ (see Sec. 3.4). By minimizing this loss, the agent learns to find $\mathbf{C}$. This is then used to assess the quality of the meta-episode $\mathbf{B}_j$.

Note that computing $\mathbf{C}$ is equivalent to finding an approximate solution to the system of linear equations $\mathbf{B}_j \mathbf{C} = \mathbf{B}$. Therefore, $g$ requiring $\mathbf{B}$ as an input is an unavoidable constraint. Consequently, $\mathbf{C}$ cannot be computed during task exploration, where the agent is limited to collecting a single meta-episode. We present an alternative approach for optimizing $f_u$ for sample-efficient exploration. Consider a target

$$\delta_j = 1 - \exp\left(-\xi \overline{(\mathbf{B}_j \mathbf{C} - \mathbf{B})^2}\right), \tag{8}$$

where $\xi$ is a constant and $\overline{\mathbf{T}}$ gives the element-wise mean of any tensor $\mathbf{T}$. This target measures the linear task reconstruction in Eq. 6, such that $\delta_j \approx 0$ implies $\mathbf{B}_j$ is a good approximation of $\mathbf{B}$. We therefore optimize $f_u$ to learn a latent space that encodes this information. Specifically, for a given $\boldsymbol{\Gamma}$, we optimize the similarity between any two episode representations $\boldsymbol{\Gamma}_{:,k}$ and $\boldsymbol{\Gamma}_{:,k'}$ to be $\delta_j$, for $k, k' \in [K]$. As in Sec. 3.2, we use cosine similarity. We can now define the loss

$$\mathcal{L}_{\text{contr}} = \mathbb{E}_{\mathcal{M}_i \sim p(\mathcal{M}),\, \mathbf{B} \sim P_{\mathcal{M}_i}^{\pi^{\text{explore}}},\, j \sim U(Q)} \left[ \operatorname{MSE}\left(S_C(\boldsymbol{\Gamma}^{\mathsf{T}}, \boldsymbol{\Gamma}),\, \boldsymbol{A}(\delta_j)\right) \right], \tag{9}$$

where $\boldsymbol{\Gamma} = f_u(\mathbf{B}_j)$,

and $\boldsymbol{A}(\delta_j) \in \mathbb{R}^{K \times K}$ is the matrix in which off-diagonal elements are $\delta_j$ and diagonal elements are 1. Eq. 9 is a form of contrastive learning for RL (Eysenbach et al., 2022; Erraqabi et al., 2022), where $\boldsymbol{\Gamma}_{:,k}$ and $\boldsymbol{\Gamma}_{:,k'}$ are pushed apart or pulled together according to a soft similarity signal $\delta_j$. This signal quantifies the similarity between the corresponding inputs $\mathbf{B}_{j,:,k}$ and $\mathbf{B}_{j,:,k'}$. Since this contrastive objective operates over cosine similarity, pushing dissimilar representations corresponds to maximizing the angle between them, while pulling corresponds to aligning similar representations by minimizing the angle. Intuitively, given the definition of similarity in Eq. 8, the angles between the $K$ columns in $\boldsymbol{\Gamma}$ are optimized to encode how well $\mathbf{B}_j$ reconstructs the task data $\mathbf{B}$. Therefore, collecting a meta-episode where all vectors in $\{\boldsymbol{\Gamma}_{:,k}\}_{k=1}^K$ are approximately orthogonal to each other is now equivalent to searching for a $\mathbf{B}_j$ that closely approximates $\mathbf{B}$. The important advantage is that this method of finding $\mathbf{B}_j$ can be used for $K$-shot adaptation. In contrast, the alternative approach of collecting $Q$ meta-episodes would be much more sample-inefficient. The encoder $f_u$ can now be used to train an exploration policy $\pi^{\text{explore}}$ to find a meta-episode $\mathbf{B}_j$, as described in Sec. 3.2.

### 3.4 Architecture and Optimization

We now introduce the architecture of our encoder $g$. Because of their close relation, we define $f_u$ to be a sub-network of $g$ (i.e., $g$ and $f_u$ are optimized together). Recall that $f_u$ separates informative meta-episodes from non-informative ones during task exploration. Moreover, $f_u$ is constrained to encode meta-episodes online, timestep by timestep, as they are being collected. To satisfy this, we model $f_u$ as a unidirectional transformer with parameters $\omega_u$ and train it using autoregressive attention masks (Vaswani et al., 2017). To obtain more informative contexts once all task data has been collected, we define an additional bidirectional transformer $f_b$ of size $d_{\text{model}}$, with parameters $\omega_b$. Finally, we set $g$ to be composed of $f_u$ and $f_b$.[3]

Given $\mathbf{B}$, $f_b$ computes two new latent representations $\boldsymbol{z}, \mathbf{Z} = f_b(\mathbf{B}; \omega_b)$. Importantly, $f_b$ is bidirectional, so it encodes data "offline", i.e., after task exploration is over. Although $\mathbf{B}$ refers to a single-task collection of meta-episodes, in practice, we meta-train on a batch of such tensors, each collected from a different task. This batching accelerates learning. More importantly, it allows us to capture the shared task structure across $p(\mathcal{M})$ by computing the vector $\boldsymbol{z} \in \mathbb{R}^{d_{\text{model}}}$ from the full batch of meta-episodes. In contrast, the tensor $\mathbf{Z} \in \mathbb{R}^{Q \times HK \times d_{\text{model}}}$ is a task-specific representation of $\mathcal{M}_i$, encoded solely from the dataset $\mathbf{B}$ collected in $\mathcal{M}_i$. We use $\mathbf{Z}_j$ to denote the matrix $\mathbf{Z}_{j,:,:} \in \mathbb{R}^{(HK) \times d_{\text{model}}}$ encoding the meta-episode $\mathbf{B}_j$, for any $j \in [Q]$. While $\mathbf{Z}$ and $\boldsymbol{\Gamma}$ have similar roles, $\boldsymbol{\Gamma}$ is optimized to learn representations that are only useful for task exploration, so it may fail to capture structure that is important for solving tasks, but irrelevant for exploration. For a given meta-episode $\mathbf{B}_j$, we compute the task context vector $\boldsymbol{c} \in \mathbb{R}^{(HKd_{\text{model}})}$ as a function of $\boldsymbol{z}$, $\mathbf{Z}_j$, and $\boldsymbol{\Gamma}$.[4] Specifically, we use a function $h_{\omega_h}$, parameterized by $\omega_h$, to encode $\mathbf{B}_j$ into

$$\boldsymbol{c} = g_\omega(\mathbf{B}_j) = h_{\omega_h}(\boldsymbol{z}, \mathbf{Z}_j, \boldsymbol{\Gamma}), \tag{10}$$

where $g_\omega$ has the concatenated parameters $\omega = \omega_u \oplus \omega_b \oplus \omega_h$. In practice, $h$ consists of two attention layers. Our encoder architecture is shown in Fig. 3.

To train our encoder $g$, we define two pre-training heads: $h_{\text{t-rec}}$ and $h_{\text{rec}}$, parameterized by $\omega_{\text{t-rec}}$ and $\omega_{\text{rec}}$, respectively. The former is used to compute the coefficient tensor $\mathbf{C} = h_{\text{t-rec}}(g_\omega(\mathbf{B}), g_\omega(\mathbf{B}_j); \omega_{\text{t-rec}})$ required in Eq. 7. That is, $h_{\text{t-rec}}$ combines the representations of $Q$ meta-episodes, $g_\omega(\mathbf{B})$, with the representation of the $j$-th meta-episode, $g_\omega(\mathbf{B}_j)$. Note that, when using $\mathbf{C}$ to compute $\delta_j$ in Eq. 8, we instead use a tensor encoded by target parameters $\hat{\omega}_{\text{t-rec}}$. To increase training stability, we only update $\hat{\omega}_{\text{t-rec}}$ to $\omega_{\text{t-rec}}$ every $\nu$ iterations Mnih et al. (2015). The latter head is used during the optimization of $g_\omega$ to reconstruct data from corrupted inputs. This is a common approach for pre-training bidirectional transformers. Since both $h_{\text{t-rec}}$ and $h_{\text{rec}}$ are only used during meta-training, they are replaced by policies $\pi^{\text{explore}}$ and $\pi$ during meta-testing. Fig. 4 gives an overview of the heads.

---

[3]Due to overlapping terminology, LaSER can be seen as performing in-context learning in two senses. From the meta-RL perspective, LaSER uses in-context policies conditioned on contextual information. From a different perspective, these context vectors are computed using transformers, which, due to their self-attention mechanism, are considered to be in-context learners.

[4]The reader might note that the $(HKd_{\text{model}})$-dimensional context vector $\boldsymbol{c}$ does not necessarily reduce the size of the input $\mathbf{B}_j$. A high dimensional $\boldsymbol{c}$ is beneficial since preserving the structure of $\mathbf{B}_j$ (i.e., the $HK$ timesteps) makes it simpler to train transformers $f_u$ and $f_b$. Despite this, $\boldsymbol{c}$ is still a useful representation of $\mathbf{B}_j$. Each of the $HK$ output timesteps has been computed from its corresponding input timestep and the rest of the input sequence. This enables dimensionality reduction later on. Specifically, we use the task policy $\pi$ to reduce $\boldsymbol{c}$ to a $d_{\text{model}}$-dimensional vector.

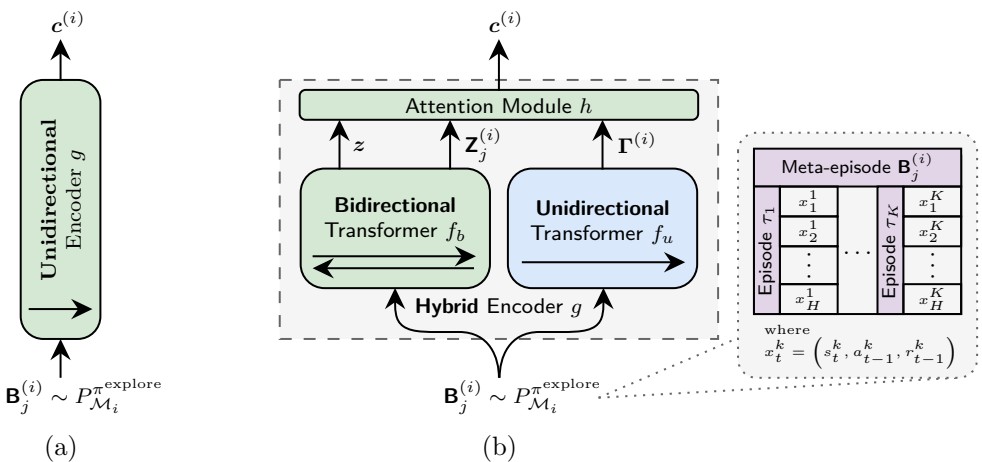

Figure 3: Two types of meta-RL encoders. (a) Unidirectional encoder, common in meta-RL. It processes task exploration data online, step by step, as it is being collected. (b) LaSER's encoder $g$. We enhance standard architectures by adding a bidirectional encoder. Encoder $g$ can be used online for task exploration or offline to compute context $c^{(i)}$. We use $\cdot^{(i)}$ to differentiate between meta-episodes and representations belonging to different tasks. Note that we define timesteps $x_t^k$ to contain the current state $s_t^k$, but the previous action $a_{t-1}^k$ and reward $r_{t-1}^k$, with $x_1^k = (s_1^k, \_, \_)$. This simplifies the training of unidirectional encoders.

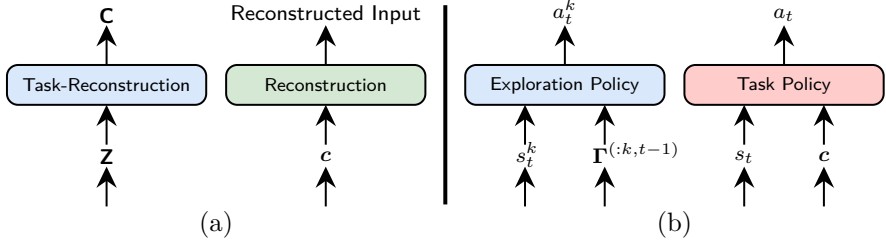

Figure 4: LaSER's pre-training and policy heads. (a) Task-reconstruction head $h_{\text{t-rec}}$ and reconstruction head $h_{\text{rec}}$, used for pre-training encoder $g$ for task exploration and task learning, respectively. (b) Policies $\pi^{\text{explore}}$ and $\pi$, used for task exploration and task solving, respectively.

We train $g_\omega$ using masked self-supervision (Devlin et al., 2019; Lewis et al., 2019). The encoder learns useful representations by adding noise to the input, reconstructing the original input, and optimizing a reconstruction loss $\mathcal{L}_{\text{rec}}$ (see Appendix B for more details). We extend this idea and define the loss of $g_\omega$ as

$$\mathcal{L}_{\text{LaSER}}(\omega, \omega_{\text{rec}}, \omega_{\text{t-rec}}) = c_{\text{rec}}\mathcal{L}_{\text{rec}}(\omega, \omega_{\text{rec}}) + c_{\text{t-rec}}\mathcal{L}_{\text{t-rec}}(\omega, \omega_{\text{t-rec}}) + c_{\text{contr}}\mathcal{L}_{\text{contr}}(\omega) + c_{\mathcal{R}}\mathcal{R}(\omega), \tag{11}$$

where $c_{\text{rec}}, c_{\text{t-rec}}, c_{\text{contr}}, c_{\mathcal{R}}$ are weighting coefficients, and $\mathcal{R}$ is a regularization term. It regularizes the latent spaces generated by $f_b$ and $f_u$. Following Piratla et al. (2020), $\mathcal{R}$ enforces a soft orthogonality constraint between the vector representations $z$ and $\mathbf{Z}_{j,t,:}$, for each $j \in [Q]$ and timestep $t \in [HK]$, by minimizing $(z^\top \mathbf{Z}_{j,t,:})^2$. Intuitively, $\mathbf{Z}$ is encouraged to avoid capturing information already contained in $z$, and instead focus on task-specific structure. Furthermore, $\mathcal{R}$ normalizes the vector representations $z$, $\mathbf{Z}_{j,t,:}$, and $\mathbf{\Gamma}_{:,k}$, for all $j \in [Q], t \in [HK], k \in [K]$.

## 3.5 Meta-Training and Meta-Testing

The LaSER meta-training algorithm is shown in Alg. 1. It contains two training phases. We first train the exploration policy $\pi_\phi^{\text{explore}}$, parameterized by $\phi$, and the encoder $g_\omega$, parameterized by $\omega$, for $N_{\text{explore}}$ iterations. Note that these components are trained together because they are interdependent. We alternate

between optimizing one while keeping the other fixed. In the second stage, these two components are fixed and only the task policy $\pi_\theta$, parameterized by $\theta$, is optimized for $N_{\text{task}}$ iterations.

---

**Algorithm 1** LaSER Meta-Training

---

    **Input** $p(\mathcal{M})$, task distribution

    **Output** $\pi_\theta$, task policy; $\pi_\phi^{\text{explore}}$, exploration policy; $g_\omega$, encoder; $\hat{\boldsymbol{z}}$, shared component

1: $\theta, \phi, \omega \leftarrow$ initialize randomly
2: **for** $n = 1, 2, \ldots, N_{\text{explore}}$ **do**
3:     $\omega \leftarrow \text{train\_encoder}(p(\mathcal{M}), \pi_\phi^{\text{explore}}, g_\omega)$                    ▷ Alg. 3
4:     $\phi \leftarrow \text{train\_exploration\_policy}(p(\mathcal{M}), \pi_\phi^{\text{explore}}, g_\omega)$       ▷ Alg. 4
5: **end for**
6: $\mathcal{B} \leftarrow \left\{ \mathbf{B} \sim P_{\mathcal{M}_i}^{\pi_\phi^{\text{explore}}} \middle| \mathcal{M}_i \sim p(\mathcal{M}) \right\}$       ▷ Collect a batch of data from multiple tasks
7: $\hat{\boldsymbol{z}}, \_\_ \leftarrow f_b(\mathcal{B}; \omega_b)$
8: **for** $n = 1, 2, \ldots, N_{\text{task}}$ **do**
9:     $\theta \leftarrow \text{train\_task\_policy}(p(\mathcal{M}), \pi_\theta, \pi_\phi^{\text{explore}}, g_\omega, \hat{\boldsymbol{z}})$       ▷ Alg. 5
10: **end for**
11: **return** $\pi_\theta, \pi_\phi^{\text{explore}}, g_\omega, \hat{\boldsymbol{z}}$

---

Recall that $\boldsymbol{z}$ is computed from multiple tasks, which is only feasible during pre-training. Therefore, we compute a fixed $\hat{\boldsymbol{z}}$ from the pre-training data and use it to train the task policy and perform meta-testing. During meta-testing, for any $\mathcal{M}_i \sim p(\mathcal{M})$, the meta-trained agent first collects $K$ episodes and computes the latent context $\boldsymbol{c}$. Then, it uses $\boldsymbol{c}$ to find the optimal policy for $\mathcal{M}_i$. This is shown in Alg. 2.

---

**Algorithm 2** LaSER Meta-Testing

---

    **Input** $p(\mathcal{M})$, task distribution; $\pi_\theta$, task policy; $\pi_\phi^{\text{explore}}$, exploration policy; $g_\omega$, encoder;
        $\hat{\boldsymbol{z}}$, shared component

1: **for** $\mathcal{M}_i \sim p(\mathcal{M})$ **do**
2:     $\mathbf{B}_j \sim P_{\mathcal{M}_i}^{\pi_\phi^{\text{explore}}}$                                 ▷ Sample exploration meta-episode
3:     $\boldsymbol{c} \leftarrow g_\omega(\mathbf{B}_j)$                     ▷ Compute $\boldsymbol{c}$, using $\hat{\boldsymbol{z}}$ as the shared component
4:     $\tau \sim P_{\mathcal{M}_i}^{\pi_\theta(a|s,\boldsymbol{c})}$
5:     Measure return in exploitation episode $\tau$
6: **end for**

---

## 4 Related Work

**Meta-RL.** The earlier successes of modern meta-RL start with MAML (Finn et al., 2017) and RL$^2$ (Duan et al., 2016; Wang et al., 2016). MAML, together with other MAML-inspired algorithms (Sung et al., 2017; Li et al., 2017; Gupta et al., 2018; Zintgraf et al., 2019), are gradient-based methods for meta-training policies that can adapt to new tasks by taking a small number of task-specific gradient steps. Gradient-based approaches usually implement an explicit dual-loop algorithm that follows the standard meta-learning paradigm of adapting to tasks in an inner loop, while meta-learning adaptation strategies in an outer loop. On the other side of the spectrum, RL$^2$ is an in-context meta-RL approach, meta-training a policy to behave like an RL algorithm that learns from collected task contexts (Laskin et al., 2023; Moeini et al., 2025). This learning is usually represented by a forward pass through the pretrained policy, with no task-specific weight updates. Recent meta-RL techniques, including ours, follow this paradigm of first identifying task dynamics, and then adapting a task-agnostic policy to them. Generally, this in-context learning tends to be more sample-efficient than gradient-based methods, which is ideal in few-shot adaptation (Beck et al., 2023b).

**In-Context Policies.** To learn task-specific policies, RL$^2$ uses a recurrent architecture that implicitly conditions the policy on task history. Later works make this conditioning more explicit (Rakelly et al.,

2019; Zhou et al., 2019; Zintgraf et al., 2019). Specifically, they show that in-context policies can be trained using standard RL optimization by simply augmenting the input state with task contexts. Similar ideas have been explored in the closely related area of unsupervised representation learning for RL (Igl et al., 2018; Papoudakis et al., 2021; Botteghi et al., 2025). This approach has since become standard in meta-RL, with subsequent research focusing more on learning informative task contexts than on novel architectures or optimization algorithms for in-context policies. Recently, Beukman et al. (2024) observed that this simple method may struggle when there is high variation across the optimal task-specific policies, which can arise in complex task distributions. As an alternative, Beck et al. (2023a) and Beukman et al. (2024) propose that meta-training hypernetworks (Ha et al., 2017) may lead to better generalization. Therefore, they introduce hypernetworks that take the task context as input and generate the weights of a context-dependent policy. While this architectural change may better leverage task contexts, their approach is still limited by the use of standard RL optimization, which constrains task space exploration. Additionally, this method inherits issues related to hypernetworks, such as slow and unstable training (Ortiz et al., 2023; Chauhan et al., 2024; Beukman et al., 2024). In contrast, our proposed meta-reward is not an adaptation of standard learning methods to meta-RL, but is explicitly designed for the meta-RL setting. Moreover, it is architecture-agnostic, introduces little overhead, and allows in-context policies to be optimized through standard RL. This may simplify solving meta-RL problems, as practitioners can rely on stable and well-understood RL algorithms.

**Exploration in Meta-RL.** A considerable body of literature also focuses on exploration in meta-RL. As opposed to standard RL, meta-RL exploration strategies can be learned from interactions with the environment and then applied to new tasks. Since identifying and solving RL tasks requires exploration, all meta-RL algorithms learn to explore, at least implicitly. However, several works have shown the benefits of explicitly learning to explore. Rakelly et al. (2019) use posterior sampling to explore. Zintgraf et al. (2021b) consider Bayes-optimal policies, which optimally trade-off exploration and exploitation, and meta-learn approximations of such policies. They later extend their work in Zintgraf et al. (2021c) by encouraging the agent to explore novel hyper-states during meta-training. Some other approaches make exploration more efficient by structuring the exploration space through contrastive learning (Fu et al., 2021; He et al., 2024; Yu et al., 2024), information gain (Liu et al., 2021; Jiang et al., 2021; Zhang et al., 2021), or task clustering (Chu et al., 2024). Gradient-based meta-RL can also explicitly learn to explore. Gupta et al. (2018) explore by adding structured, meta-learned noise to the policy. Similarly, Stadie et al. (2018) enhance MAML and $RL^2$ by adding an exploration term to the meta-RL objective. Gurumurthy et al. (2020) add self-supervised objectives to lower variance during exploration. Finally, several of the works discussed improve exploration even further by decoupling the exploration and task-solving policies (Zhou et al., 2019; Gurumurthy et al., 2020; Liu et al., 2021; Fu et al., 2021; Norman & Clune, 2024). LaSER also collects data using a decoupled exploration policy, and then meta-learns a structured exploration space. However, it uses a novel objective that encourages the collection of a single meta-episode which serves as a low-rank linear representation of a larger dataset drawn from the same task. An important distinction to numerous previous works is that task exploration depends only on the structure of the data, while being agnostic to the RL objective of the task policy. This may be advantageous for out-of-distribution adaptation, where meta-test tasks may have goals that differ from those seen in meta-train tasks. Therefore, data collected for maximizing meta-training return might not always be relevant during meta-testing.

**Meta-Learning Contexts.** Once task data has been collected, a straightforward way to obtain informative contexts is through recurrent neural networks (Duan et al., 2016; Wang et al., 2016). More sophisticated methods include meta-learning latent representations of value functions (Rakelly et al., 2019) or MDP dynamics (Zhou et al., 2019; Zintgraf et al., 2021b;c), model-based meta-RL (Clavera et al., 2018; Nagabandi et al., 2018), hybrid techniques that combine in-context and gradient-based methods (Imagawa et al., 2022), using permutation variant and invariant sequence models (Beck et al., 2024), or enhancing tasks by incorporating language instructions (Bing et al., 2023). Attention mechanisms (Bahdanau et al., 2015) and transformers (Vaswani et al., 2017) have also been adopted by the meta-RL community. Their success in in-context learning, long-sequence modeling, and efficient parallelizable training aligns well with the needs of in-context meta-RL. Earlier research focused solely on meta-learning through attention (Mishra et al., 2018), while more recent work used transformer architectures (Melo, 2022; Shala et al., 2024). A limitation of previous works that attempt to meta-learn task dynamics is that only unidirectional encoders are used. This constraint arises naturally since task exploration and task learning are coupled. Specifically,

task data must be encoded online while it is being collected, in order to guide the exploration policy at the next timestep. An obvious limitation is that only interactions between the current and past timesteps are considered. LaSER improves this design by using an additional bidirectional encoder that also considers future timesteps, which could possess useful information and lead to richer representations (Devlin et al., 2019; Banino et al., 2022). While the aforementioned constraint cannot be avoided during task exploration, it need not restrict the computation of the final task context once all exploration data has been collected.

# 5 Experiments

In this section, we present empirical results for LaSER. We first introduce the environment and algorithms we use in Sec. 5.1. In Sec. 5.2, we compare LaSER with other types of meta-RL algorithms. Next, we perform two ablation studies by analyzing individual stages of the meta-RL pipeline (Fig. 1). In Sec. 5.3 we evaluate our novel approach to meta-training in-context task policies. This corresponds to LaSER's task-solving phase, which we perform using ground-truth contexts instead of task exploration and learning. Finally, we assess LaSER's task exploration and task learning stages in Sec. 5.4 by visualizing the latent task contexts computed during meta-testing.

## 5.1 Experimental Setup

### 5.1.1 Environments

We evaluate LaSER on the meta-RL benchmark MEWA (Stoican et al., 2023).[5] MEWA provides a distribution of tasks that share the same central idea: certain states, called critical states, can lead to mistakes. These mistakes affect the final return negatively. The probability of a mistake happening depends on both the type of critical state and the dynamics of the task. By exploring and learning a task, RL agents must find the optimal policy for avoiding high-risk mistakes while minimizing task delay. Importantly, Stoican et al. (2023) ensure MEWA's task distribution has no globally optimal policy (i.e., no policy can zero-shot solve all tasks). So, an agent can only be optimal by collecting new data and adapting. This property makes MEWA ideal for our case, as it allows us to test LaSER's ability to explore tasks.

MEWA evaluates agents on their ability to take optimal actions in different types of "critical" states. These critical states provide agents with two options. Consider a critical state $s_x$ of type $x$. The agent's first option is to take a "risky" action. This may lead to a mistake of type $x$ happening, which in turn leads to a large delay in task completion. The probability of a type $x$ mistake happening depends on the transition function of the task. The second option is a "safe" action. This provides a guaranteed small delay and leads to a state $s_{x-1}$ of type $x$. An important feature shared by all MEWA tasks is that for any $y < x$, mistakes of type $y$ are less likely to happen than mistakes of type $x$. Therefore, depending on the task, taking several safe actions before a risky action could be optimal.

We meta-train all meta-RL agents on the narrow task distribution analyzed by Stoican et al. (2023). This corresponds to a distribution $p(\mathcal{M})$ in which any task $\mathcal{M}_i \sim p(\mathcal{M})$ has four different types of critical states and can be described by a vector $\boldsymbol{p}^{(i)} \in [0,1]^4$. For each task $\mathcal{M}_i$, the probability of making a mistake of type $x \in [4]$ is $\boldsymbol{p}_x^{(i)} \sim \mathcal{N}(\boldsymbol{\mu}_x, 0.12)$, where $\mathcal{N}$ is a normal distribution and $\boldsymbol{\mu}^\mathsf{T} = [0.38, 0.28, 0.19, 0.09]$. Note that we clip all $\boldsymbol{p}_x^{(i)}$ to $[0,1]$. See Appendix E for a more formal definition of MEWA's tasks.

We meta-test on the fixed set of 12 tasks proposed by Stoican et al. (2023). Their corresponding $\boldsymbol{p}^{(i)}$ vectors are listed in Appendix E, Table 3. We assign a score to each task, given by the average probability of a mistake of any type happening, i.e., $1/4 \sum_{x=1}^{4} \boldsymbol{p}_x^{(i)}$. We use this as a rough estimate of the similarity between tasks, such that tasks with a similar score require similar (but not identical) optimal policies. Additionally, note that half of the meta-testing tasks are out-of-distribution (OOD) tasks. OOD tasks lie outside the meta-training distribution $p(\mathcal{M})$, so they are more challenging to explore and solve during meta-testing.

---

[5]We use the publicly available code for the MEWA benchmark at https://github.com/RStoican/MEWA.

Note that no globally optimal non-adaptive policy exists for these 12 meta-testing tasks. Additionally, the difference in performance between the optimal non-adaptive and optimal adaptive policies is high enough to evaluate the agent's ability to generalize and adapt.

MEWA offers 3 baselines to compare our agents to during meta-testing. The *random* baseline represents the expected performance of an agent that takes uniformly random actions. The *task-agnostic* baseline represents an agent that always takes optimal actions in MEWA's non-critical states. These are states in which the optimal action can be computed even if the policy has no task information, i.e., for the optimal value function $V^*$, if $V^*(s, c) \leq V^*(s)$ for all $c$, then $s$ is a non-critical state. For any critical state, the *task-agnostic* baseline considers all actions that could potentially be optimal in a valid task, then takes one at random. Finally, the *optimal* baseline shows the expected return of the optimal meta-RL agent, i.e., the agent that adapts to and solves each task optimally.

### 5.1.2 Algorithms

We compare our method, LaSER, with four other meta-RL algorithms, MAML (Finn et al., 2017), PEARL (Rakelly et al., 2019), VariBAD (Zintgraf et al., 2021b), and DREAM (Liu et al., 2021). We meta-train each algorithm on $3,000$ tasks sampled from MEWA's meta-training distribution (see Sec. 5.1.1). For each task $\mathcal{M}_i$, the agents are allowed to collect $K = 4$ episodes during the task exploration phase. Their performance on $\mathcal{M}_i$ is then given by a single exploitation episode collected by the task policy. Besides this final return, we also analyze an agent's ability to improve its performance as more data is collected. All tasks have a horizon of $H = 50$.

LaSER's hyperparameters are listed in Appendix H. We selected these based on performance on MEWA's meta-training task distribution. For implementation details and hyperparameters for MAML, PEARL, VariBAD, and DREAM, see Appendix F.

To ensure a fair evaluation, we first show that all algorithms converge during meta-training. For this, we evaluate agents on MEWA's meta-training distribution. As shown in Fig. 5, LaSER's exploration policy, encoder, and task policy converge across 10 seeds. The encoder loss is computed by Eq. 11, with Fig. 9 showing results for each of its terms. The exploration return is computed using intrinsic rewards (see Eq. 5) collected by the exploration policy. We first meta-train $\pi_\phi^{\text{explore}}$ and $g_\omega$ for 220 million timesteps, and then $\pi_\theta$ for 7.5 million timesteps. Note, however, that for the first 50 million timesteps we only train $f_u$ (and thus $g_\omega$), without updating $\pi_\phi^{\text{explore}}$ since its training is conditioned on representations learned by $f_u$. For these initial updates, we use data collected by a uniformly random policy. Finally, Appendix G provides convergence results for all the baseline algorithms. All their task policies converge (Fig. 10), together with PEARL's, VariBAD's, and DREAM's encoders (Fig. 11), and DREAM's exploration policy (Fig. 12). Note that MAML does not have an encoder, and DREAM is the only baseline algorithm with a separate exploration policy.

We also analyze the computational complexity of meta-training LaSER. Specifically, we measure the wall clock time required to meta-train all three components of LaSER, averaging results over 10 seeds. We report both the total meta-training time and the average time per timestep. For total time, we stop training once performance (i.e., return or loss) stops improving in all seeds. We provide detailed measurements in Fig. 13.

In terms of total time, LaSER is the most expensive, being 1.34 times slower than the next slowest algorithm, DREAM. This is followed by PEARL, and finally by MAML and VariBAD. Most of LaSER's time complexity comes from its transformer encoders. While transformers are powerful, they are also known to demand a high amount of data. However, when considering the average time per timestep, LaSER is more efficient. Specifically, LaSER is 11.93 and 5.86 times faster than DREAM and PEARL, respectively. Furthermore, it is only 1.71 and 2.64 times slower than MAML and VariBAD, respectively.

### 5.2 Overall Performance

We first evaluate LaSER's ability to solve MEWA's meta-testing tasks. For each algorithm, we meta-test the agent obtained at the end of meta-training. Note that before being evaluated on a task, each agent is allowed to explore for $K = 4$ episodes. We report average returns over 10 seeds in Fig. 6 and Table 1. Because of the inherent randomness in MEWA's tasks, we take an additional step to ensure the results are not due to

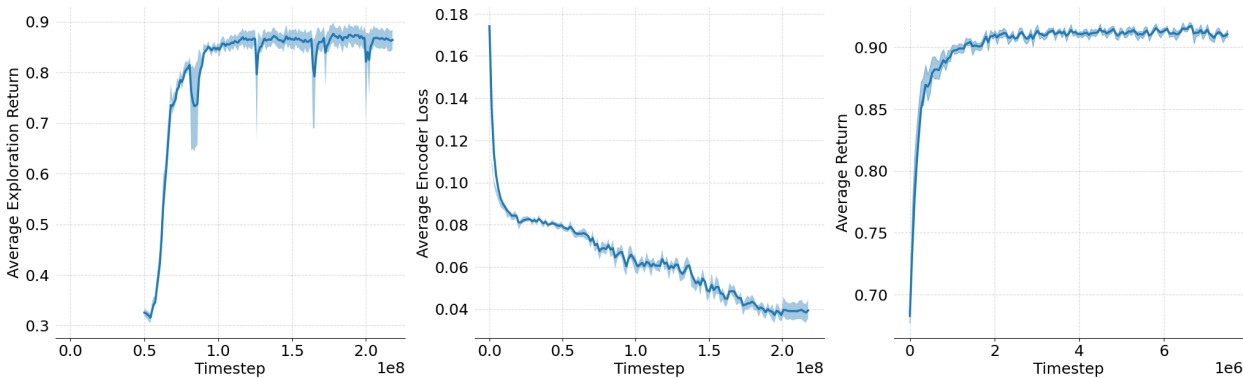

Figure 5: Performance on tasks from MEWA's meta-training task distribution. For each metric, the shaded areas report the standard error of the average return across 10 random seeds. We use exponential moving average (EMA) smoothing $y_j = \alpha_{\text{EMA}} x_j + (1 - \alpha_{\text{EMA}}) y_{j-1}$ for each point $x_j$, with $\alpha_{\text{EMA}} = 0.6$. We first meta-train the exploration policy and the encoders for $2.2e8$ timesteps, then the task policy for $7.5e6$. Additionally, we only start exploring tasks after 50 million timesteps. (left) Undiscounted exploration return of policy $\pi_\phi^{\text{explore}}$ for exploration reward function $\tilde{R}$. (middle) Encoder loss $\mathcal{L}_{\text{LaSER}}$. (right) Undiscounted return of policy $\pi_\theta$.

random chance. For each seed, we collect 180 exploitation episodes per agent on each of the 12 meta-testing tasks and average over all their returns. Finally, we also report per-task performance in Fig. 14.

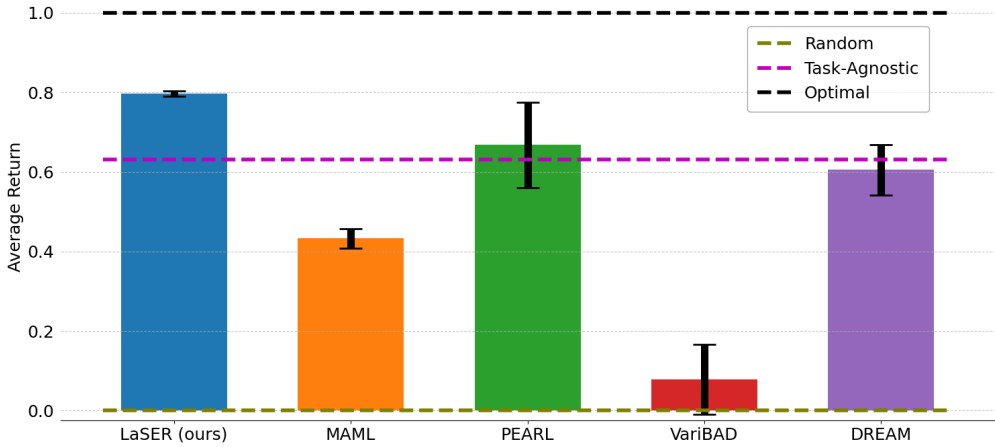

Figure 6: Average returns achieved on MEWA's meta-testing tasks. Each agent is meta-tested with an exploration budget of $K = 4$. Results are averaged over 10 seeds, with error bars indicating standard error. Note that we normalize results between the *random* and *optimal* baselines.

| LaSER (ours) | MAML | PEARL | VariBAD | DREAM |
|---|---|---|---|---|
| $\underline{0.8 \pm 0.021}$ | $0.43 \pm 0.072$ | $0.67 \pm 0.32$ | $0.08 \pm 0.262$ | $0.6 \pm 0.191$ |

Table 1: MEWA returns averaged across 10 seeds ($\pm$ std).

LaSER achieves the highest average returns among all meta-RL algorithms. It also proves to be the most stable algorithm, with a post-adaptation standard deviation of just 0.02 across all seeds. LaSER is one of only two methods to outperform the *task-agnostic* baseline. The other algorithms only outperform the *random* baseline. PEARL learns a strong global policy, but is less stable and underperforms compared to

LaSER. While DREAM appears more stable, its performance is slightly below the *task-agnostic* baseline. In contrast, MAML and VariBAD perform poorly, with VariBAD being both unstable and close in performance to the *random* baseline.

### 5.3 Solving Tasks

As an ablation study, we evaluate our approach for meta-training in-context task policies, which was introduced in Sec. 3.1. The results in this section correspond solely to the task-solving phase of Fig. 1. We meta-test task policies by rolling them out before and after they receive a task context $c$. Our primary objective is to assess adaptability, i.e., a policy's ability to use $c$ to improve its performance. We quantify this as the difference between post-adaptation and pre-adaptation return.

We make two important simplifications to the standard MEWA benchmark.

- To ensure a fair comparison of task policies alone, we meta-train and meta-test without task exploration or task learning. Instead, for each task, we use a ground truth, oracle-provided context vector $c$, which is normally unavailable to the agent.

- To assess adaptability to non-stationary dynamics, we use a simpler, multi-task objective: we meta-train and meta-test on MEWA's 12 meta-testing tasks.

Despite these restrictions, MEWA's guarantee for the nonexistence of a globally optimal policy still holds. That is, only adaptive policies can achieve maximum return.

LaSER combines PPO and the proposed meta-reward $r_t^+$ (see Eq. 2) to optimize the in-context policy $\pi_\theta$. We compare it to the simpler approach of using in-context policies optimized through standard PPO, as well as to Decision Adapters (Beukman et al., 2024, DAs), which use hypernetworks to generate task-specific policies.[6] For a fair comparison, the in-context PPO policy has the same architecture as LaSER's policy. We also use PPO to optimize DAs, so the main difference between these and the other two methods is the architectural change. Appendix F.4 provides implementation details and hyperparameters for DAs.

The meta-training results are shown in Fig. 7 and Table 2. Our method achieves the highest average return, while also stabilizing meta-training. It is also the only method to find a policy that adapts optimally, which occurred in 5/10 seeds. In-context PPO is slightly less stable, presumably because the agent is trying to find a single policy that maximizes returns across all tasks. However, without task-specific adaptation, it fails to find such a globally optimal policy. Hypernetworks, on the other hand, are highly unstable. This is likely due to hypernetworks being inherently challenging to train (Ortiz et al., 2023; Chauhan et al., 2024; Beukman et al., 2024). They also achieve the lowest average return.

|  | LaSER Task Policy (ours) | In-Context PPO | Decision Adapter Hypernetwork |
|---|---|---|---|
| Return | $\underline{0.97} \pm \underline{0.006}$ | $0.93 \pm 0.014$ | $0.86 \pm 0.149$ |
| Delta Return | $\underline{0.47} \pm \underline{0.019}$ | $0.07 \pm 0.067$ | $0.12 \pm 0.164$ |

Table 2: Average returns and delta returns over the last $1e6$ timesteps, averaged across 10 seeds ($\pm$ std). We meta-train and meta-test on the same 12 MEWA tasks, and use oracle-provided contexts vectors $c$.

The LaSER task policy also achieves greater task-adaptation success than the other two methods. After the meta-exploration phase, the agent stabilizes towards the end of meta-training. The task-conditioned policy then consistently outperforms the pre-adaptation policy. Hypernetwork-generated policies adapt better than in-context PPO but are also more unstable. Additionally, their low average return limits the benefits of this adaptability. In-context PPO does not explicitly optimize for adaptation, so its adaptability is approximately five times lower than our method.

---

[6]Since Beukman et al. (2024) build and evaluate their DAs with ground-truth contexts in mind, their algorithm is agnostic to the context-learning mechanism, and, as reported by the authors, sensitive to noisy contexts. We therefore only consider DAs as baselines for LaSER's task policy, and not for the entire LaSER algorithm.

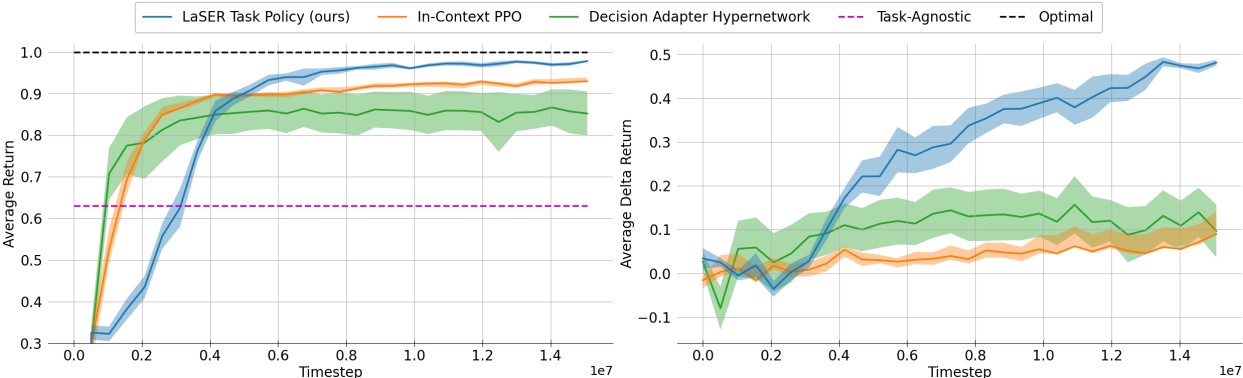

Figure 7: Task policy meta-training. We use oracle-provided task context $\boldsymbol{c}$, and meta-train and meta-test on the same set of 12 MEWA tasks. The shaded areas show the standard error of the average return across 10 random seeds. (left) Post-adaptation return of task policy $\pi_\theta$, given task context $\boldsymbol{c}$. All returns are normalized between MEWA's *random* and *optimal* baselines. To improve readability, we restrict the plot to the $[0.3, 1.0]$ range and exclude the *random* baseline (which is positioned at 0). (right) Adaptability, quantified as the difference in return achieved by $\pi_\theta$ with and without access to $\boldsymbol{c}$.

Note that our method learns more slowly initially. Compared to in-context PPO and hypernetworks, it requires more samples to pass the average return thresholds of 0.9 (for PPO) and 0.85 (for hypernetworks). We attribute this to additional meta-exploration in the state space, performed in the context of the task space. The meta-reward $r_t^+$ encourages the policy to learn how to take in-context actions that increase the gap between $V(s_t)$ and $V(s_t, \boldsymbol{c})$, for a state $s_t$ and context $\boldsymbol{c}$. This is in addition to the meta-exploration required to only maximize the standard RL objective $V(s_t)$. We hypothesize that in-context PPO and hypernetworks cannot surpass our method due to their lack of explicit task-space meta-exploration. The policies they produce optimize standard RL objectives, so they meta-explore accordingly.

Finally, we observe that our method is nearly as computationally efficient as in-context PPO and much faster than hypernetworks. We measure the average wall-clock time to roll out and optimize $\pi_\theta$, per iteration. On average, hypernetwork-based agents are approximately $3.3\times$ slower than ours, while our method is only $1.3\times$ slower than in-context PPO. See Fig. 15 for a detailed comparison. We discuss how meta-rewards cause this additional overhead in Appendix A.2.

## 5.4   Exploring and Learning Tasks

To better understand LaSER's ability to explore and encode tasks during meta-testing, we visualize its latent task space. We compare the task contexts computed by LaSER, PEARL, VariBAD, and DREAM for each of the 12 meta-testing tasks in MEWA. For each algorithm, we show results for only one of the meta-training seeds. However, we obtain similar representations for the other seeds. We first roll out the corresponding agent's meta-trained policy, collecting 100 meta-episodes per task, with $K = 4$ episodes per meta-episode. Note that in the case of LaSER, we roll out the exploration policy $\pi_\phi^{\text{explore}}$. We then encode each meta-episode $\mathcal{D}^{(K)}$ into a latent task context vector $\boldsymbol{c}$ using the agent's meta-trained encoder. Similarly, DREAM collects data using its exploration policy. We visualize two-dimensional projections of task contexts, computed using t-SNE (Van der Maaten & Hinton, 2008), in Fig. 8. The 12 tasks are sorted by similarity, i.e., average mistake probability, as in Table 3.

For LaSER, we can find clusters of task contexts. Specifically, meta-episodes collected from the first five tasks, the next four tasks, and the final three tasks appear to be projected apart from each other. We note that the tasks in each cluster share common traits. In the first cluster, mistakes of type $x \in \{3, 4\}$ never occur. In the third cluster, $\boldsymbol{p}_x^{(i)} \geq 0.9$ for any $x \neq 4$. The second cluster's average mistake probabilities lie between those of the other two clusters. The separation appears to follow this trend. More details on the tasks' properties are shown in Table 3.

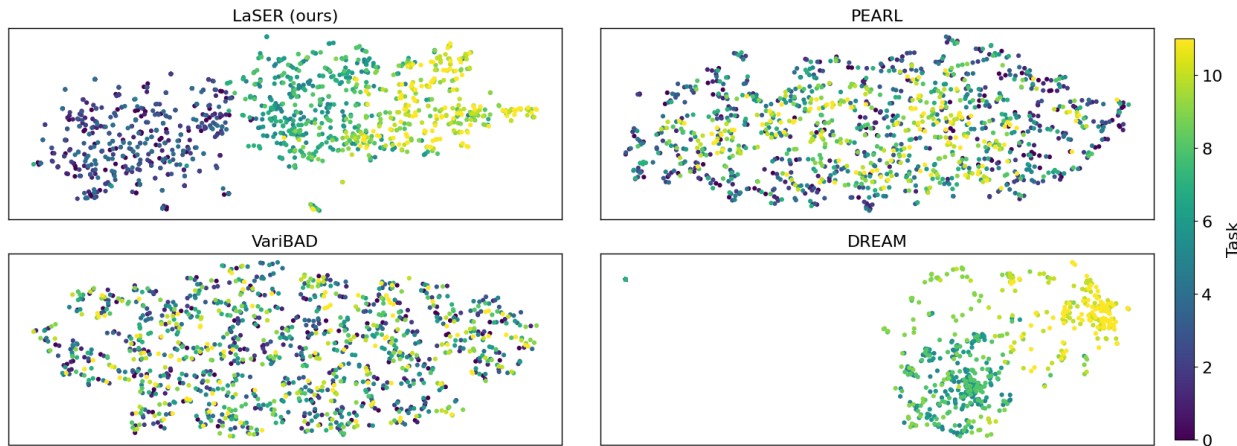

Figure 8: Latent representations of meta-episodes collected and encoded by meta-trained agents. Each of the 1200 points is a context vector belonging to one of the 12 meta-testing tasks in MEWA. The two-dimensional visualizations are computed using t-SNE. Different tasks have different colors.

In contrast, PEARL's and VariBAD's latent contexts lack clear clustering that could distinguish data collected from different tasks. This low separation may make it more difficult to identify task-specific features. DREAM, on the other hand, appears to follow a similar three-cluster separation as LaSER. A distinguishable feature is that data collected from any of the first five tasks collapses into a single context representation.

We argue that learning higher-quality clusters is not trivial. The difficulty may come from the high variance in episodes collected from the most difficult tasks. These tasks limit the exploration policy's control over the states it encounters. Additionally, the low adaptation budget of $K$ can make it difficult to separate contexts of similar tasks.

## 6 Conclusion

We introduced LaSER, a new approach for meta-learning RL exploration. Our results demonstrate that LaSER outperforms previous meta-RL algorithms on the MEWA benchmark. They also show that LaSER can meta-learn better task clustering during exploration. Additionally, we propose a novel meta-exploration bonus for training task policies efficiently. This outperforms previous approaches of meta-training in-context policies in both accumulated rewards and adaptability. Because our task-solving method is agnostic to how tasks are explored or represented, it can also be integrated into any in-context meta-RL algorithm, and then optimized using standard RL.

An important strength of LaSER is that, in the setting where contexts are meta-learned, it outperforms other algorithms, despite none of the methods being optimally adaptive. We argue that this is a result of meta-training with higher-quality task contexts. Note that the main role of these contexts remains to provide task information for adaptation during meta-testing. However, our results suggest they may also enhance performance and sample efficiency during meta-training. We believe this secondary role should also be investigated in future work.

LaSER is built upon the unique properties and requirements of the meta-RL framework, which we approach through the lens of few-shot adaptation. As a result, we tackle challenges that are specific to meta-RL, i.e., challenges that might not exist in the broader fields of RL or meta-learning. For example, our exploration algorithm leverages a key assumption about the structure of data collected in few-shot RL environments. Similarly, LaSER's task policy is meta-trained with the explicit goal that in-context policies must outperform context-agnostic policies in a meta-RL setting. This idea, realized as a form of extended meta-exploration over the task context space, leads to almost-optimal adaptive policies when ground-truth contexts are available. We hope our findings inspire future research to leverage the meta-RL framework in novel ways.

While LaSER outperforms other meta-RL algorithms, it still struggles to adapt to new tasks when task contexts are meta-learned. We discuss this from the perspective of LaSER's three main components: exploration policy, encoder, and task policy. Empirical results suggest that the first two components are well-optimized for exploring and learning tasks effectively. For instance, the encoder learned to separate MEWA's meta-testing tasks into three groups. This suggests that LaSER should be able to outperform its pre-adaptation policy in these tasks. Furthermore, our results show that the task policy can become more adaptive when restricting the meta-training task distribution and using ground truth contexts. We therefore propose the hypothesis that effectively combining these three components is non-trivial. Additionally, we suggest that future research should seek to explore and understand this issue.

### Acknowledgments

Acknowledgments will be added once the paper is accepted.

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

## A  Policy Optimization in Meta-RL

LaSER uses the proximal policy optimization (PPO) algorithm (Schulman et al., 2017) to optimize both its task-exploration and task-solving policies. Therefore, we provide a short technical description of PPO in Appendix A.1. Then, in Appendix A.2, we show a detailed overview of how PPO can be used in meta-RL, by implementing the method proposed in Sec. 3.1.

### A.1  PPO Background

For a policy $\pi_\theta$ with parameters $\theta$, Schulman et al. (2017) propose the objective

$$\mathcal{L}_t^{\text{PPO}}(\theta) = \mathbb{E}_t \left[ \mathcal{L}_t^{\text{CLIP}}(\theta) - c_1 \mathcal{L}_t^{\text{VF}}(\theta) + c_2 S[\pi_\theta](s_t, \boldsymbol{c}) \right], \tag{12}$$

where $c_1, c_2$ are constants, $\mathcal{L}_t^{\text{CLIP}}$ is the main PPO objective, and $\mathcal{L}_t^{\text{VF}}$ and $S[\pi_\theta]$ are additional objectives. $\mathcal{L}_t^{\text{VF}}$ is a squared-error loss on the value function $V_\theta(s_t, \boldsymbol{c})$, while $S[\pi_\theta]$ is the policy entropy. Note that the sole change from the original description of PPO is that $\pi_\theta(a_t \mid s_t, \boldsymbol{c})$ and $V_\theta(s_t, \boldsymbol{c})$ depend not only on the state $s_t$, but also on the task context $\boldsymbol{c}$. The clipped surrogate objective $\mathcal{L}_t^{\text{CLIP}}$ is defined as

$$\mathcal{L}_t^{\text{CLIP}} = \mathbb{E}_t \left[ \min \left( r_t(\theta) \hat{A}_t, \text{clip}(r_t(\theta), 1 - \epsilon, 1 + \epsilon) \hat{A}_t \right) \right], \tag{13}$$

for a constant $\epsilon$, policy probability ratio $r_t(\theta)$, and estimated advantage $\hat{A}_t$. The probability ratio $r_t(\theta) = \frac{\pi_\theta(a_t|s_t,\boldsymbol{c})}{\pi_{\theta_{\text{old}}}(a_t|s_t,\boldsymbol{c})}$ is computed using the parameters $\theta_{\text{old}}$ from before an update. Eq. 13 encourages small, stable policy updates that keep $\pi_\theta$ close to $\pi_{\theta_{\text{old}}}$, by constraining $r_t(\theta)$ to remain within $[1 - \epsilon, 1 + \epsilon]$.

A popular choice for the advantage estimator $\hat{A}_t$ is the generalized advantage estimator (GAE) (Schulman et al., 2015). In the meta-RL setting, the GAE can be defined as $\hat{A}_t^{\text{GAE}} = \sum_{l=0}^{H-t} (\gamma\lambda)^l \delta_{t+l}^V$ for an episode $\tau$ with horizon $H$. Here, $\gamma \in [0, 1]$ is the MDP's discount factor, $\lambda \in [0, 1]$ is an additional discount, and $\delta_t^V = r_t + \gamma V(s_{t+1}, \boldsymbol{c}) - V(s_t, \boldsymbol{c})$ is the temporal-difference (TD) error at timestep $t$ for a reward $r_t$.

### A.2  PPO for Meta-RL

In Sec. 3.1, we propose a simple change to the policy optimization objective. By replacing the environment reward $r_t$ with our proposed meta-reward $r_t^+$, PPO can be used to optimize $\pi_\theta$ to solve meta-RL tasks. That is, we compute the TD error

$$\delta_t^V(\theta) = r_t + \beta\, w(s_t, \boldsymbol{c})\, S[\pi_\theta](s_t, \boldsymbol{c}) + \gamma V(s_{t+1}, \boldsymbol{c}) - V(s_t, \boldsymbol{c}), \tag{14}$$

where $w(s_t, \boldsymbol{c}) = \max(0, \tanh(V(s_t) - V(s_t, \boldsymbol{c}) - \zeta))$. The estimated advantages can then be computed as $\hat{A}_t^{\text{GAE}}(\theta) = \sum_{l=0}^{H-t} (\gamma\lambda)^l \delta_{t+l}^V(\theta)$. Note that, because $\hat{A}_t^{\text{GAE}}(\theta)$ is now a function of the parameters $\theta$, the advantages must be recomputed every time $\theta$ is updated, i.e., after each PPO minibatch update. This is in contrast to standard PPO, where $\hat{A}_t^{\text{GAE}}$ is computed only once, before an update, and then kept fixed until new data is collected.

Intuitively, $w(s_t, \boldsymbol{c})$ measures how effective the policy $\pi_\theta$ is at using the context $\boldsymbol{c}$, for each timestep $t$. The assumption is that, while PPO-optimized policies in single-MDP settings might explore the state space sufficiently, there is no guarantee that the task context space is explored enough in a meta-RL setting. By introducing $w(s_t, \boldsymbol{c})$, whenever $\boldsymbol{c}$ does not lead to an improvement above a threshold $\zeta$, the policy is urged to explore from state $s_t$, thus learning more about $\boldsymbol{c}$. As the agent improves at using $\boldsymbol{c}$ to maximize return, the exploration bonus $w(s_t, \boldsymbol{c}) S[\pi_\theta](s_t, \boldsymbol{c})$ at state $s_t$ decreases. The standard PPO objective is only recovered when $V(s_t) \leq V(s_t, \boldsymbol{c}) + \zeta$. This signals that the agent has learned how to use $\boldsymbol{c}$ in state $s_t$, and no additional exploration is required.

## B  Masked Self-Supervised Training

The bidirectional transformer encoder $f_b$, introduced in Sec. 3, learns useful data representations by learning to reconstruct its input. Since reconstructing a meta-episode $\mathbf{B}_j$ is trivial when the entire $\mathbf{B}_j$ is given as input,

we use masked self-supervised training. We create a masked meta-episode $\mathbf{B}_j^{\text{masked}}$ by applying a stochastic masking function, similarly to Devlin et al. (2019) and Lewis et al. (2019). More precisely, $\mathbf{B}_j^{\text{masked}}$ is identical to $\mathbf{B}_j$, with the exception that each timestep $(s, a, r)$ in $\mathbf{B}_j^{\text{masked}}$ has a 15% chance of being corrupted. The loss $\mathcal{L}_{\text{rec}}$ measures the ability of $f_b$ to predict the true value of each corrupted timestep. The encoder must learn to compute these values from the non-corrupted timesteps in the input. We consider two types of corruption.

- **Masking**: with a probability of 80%, the selected timestep is replaced by a special $\langle MASK \rangle$ timestep, which carries no information.

- **Replacing**: with a probability of 10%, the selected timestep is replaced by another timestep from the same meta-episode.

The rest (i.e., 10%) of the selected timesteps are not corrupted but are still used when computing $\mathcal{L}_{\text{rec}}$. To optimize this loss, $f_b$ must learn general temporal relationships between the timesteps in a meta-episode. We compute $\mathcal{L}_{\text{rec}}$ as the MSE between the unmasked tokens in $\mathbf{B}_j$ and the reconstructed meta-episode $\mathbf{B}_j' = h_{\text{rec}}(\boldsymbol{c}; \omega_{\text{rec}})$, where $\boldsymbol{c} = g_\omega(\mathbf{B}_j^{\text{masked}})$. Note that masked meta-episodes are only used to train the encoder. When training the task policy $\pi_\theta$ or during meta-testing, we compute $\boldsymbol{c}$ using unmasked meta-episodes.

## C   Algorithms

For the sake of completeness, but also to enable reproducibility and further analysis (Phuong & Hutter, 2022), we provide pseudocode for our proposed algorithm LaSER. Alg. 1 shows our meta-training process. It is composed of three main parts. Each part optimizes one of the three LaSER components: the encoder $g_\omega$ (Alg. 3), the exploration policy $\pi_\phi^{\text{explore}}$ (Alg. 4), and the task policy $\pi_\theta$ (Alg. 5). Finally, we provide details on how we meta-test a fully trained LaSER agent in Alg. 2.

---

**Algorithm 3** train_encoder()

> **Input** $p(\mathcal{M})$, task distribution; $\pi_\phi^{\text{explore}}$, exploration policy; $g_\omega$, encoder
> **Output** $\omega$, updated parameters

1: $\mathcal{B} \leftarrow \left\{ \mathbf{B} \sim P_{\mathcal{M}_i}^{\pi_\phi^{\text{explore}}} \middle| \mathcal{M}_i \sim p(\mathcal{M}) \right\}$          ▷ Collect a batch of data from multiple tasks
2: **for** $j = 1, 2, \ldots, N_{\text{encoder}}$ **do**
3:     **for** $\mathbf{B} \in \mathcal{B}$ **do**
4:         $\mathbf{B}^{\text{masked}} \leftarrow \text{mask}(\mathbf{B})$
5:         **for** $j \in [Q]$ **do**
6:             $\boldsymbol{c} \leftarrow g_\omega(\mathbf{B}_j^{\text{masked}})$
7:             Compute reconstructed input $\mathbf{B}_j'$ as $h_{\text{rec}}(\boldsymbol{c}; \omega_{\text{rec}})$
8:         **end for**
9:         $j \sim U(Q)$
10:        $\mathbf{C} \leftarrow h_{\text{t-rec}}(g_\omega(\mathbf{B}), g_\omega(\mathbf{B}_j); \omega_{\text{t-rec}})$
11:        $\delta_j \leftarrow$ use $\mathbf{B}, \mathbf{B}_j$, and $\mathbf{C}$ in Eq. 8
12:     **end for**
13:     $\mathcal{L}_{\text{LaSER}}(\omega, \omega_{\text{rec}}, \omega_{\text{t-rec}}) \leftarrow$ compute using Eq. 7, 9, 11
14:     $\omega, \omega_{\text{rec}}, \omega_{\text{t-rec}} \leftarrow$ update using $\nabla \mathcal{L}_{\text{LaSER}}$
15: **end for**
16: **return** $\omega$

---

## D   Implementation Details

We provide additional details on our implementation of the LaSER algorithm and our architecture.

---

**Algorithm 4** train_exploration_policy()

    **Input** $p(\mathcal{M})$, task distribution; $\pi_\phi^{\text{explore}}$, exploration policy; $g_\omega$, encoder
    **Output** $\phi$, updated parameters

1:  $\mathcal{D} \leftarrow []$
2:  **for** $\mathcal{M}_i \sim p(\mathcal{M})$ **do**
3:     **for** $k = 1, 2, \ldots, K$ **do**
4:        $\mathcal{D}^{(K)} \leftarrow []$
5:        **for** $t = 1, 2, \ldots, H$ **do**
6:           $a_t^k \sim \pi_\phi^{\text{explore}}\left(\cdot \mid s_t^k, \boldsymbol{\Gamma}^{(:k,:t-1)}\right)$
7:           $\mathcal{D}^{(:k,:t)} \leftarrow (s_t^k, a_t^k, r_t^k)$
8:           $\boldsymbol{\Gamma}^{(:k,:t)} \leftarrow f_u\left(\mathcal{D}^{(:k,:t)}; \; \omega_u\right)$
9:           Collect $s_{t+1}^k, r_{t+1}^k$ by taking action $a_t^k$ in $\mathcal{M}_i$
10:       **end for**
11:     **end for**
12:     $\boldsymbol{\Gamma} \leftarrow f_u(\mathcal{D}^{(K)}; \omega_u); \qquad \boldsymbol{d} \leftarrow \frac{1}{K} S_C(\boldsymbol{\Gamma}^\mathsf{T}, \boldsymbol{\Gamma})^\mathsf{T} \mathbf{1}$
13:     Compute $\tilde{R}_k(s_t^k, a_t^k)$ using Eq. 5 for each $k, t$
14:     Replace environment rewards in $\mathcal{D}^{(K)}$ with the corresponding $\tilde{R}_k(s_t, a_t)$
15:     $\mathcal{D} \leftarrow [\mathcal{D}, \mathcal{D}^{(K)}]$
16: **end for**
17: **for** $j = 1, 2, \ldots, N_{\text{PPO}}$ **do**
18:     Optimize $\phi$ on transitions from $\mathcal{D}$ using PPO
19: **end for**
20: **return** $\phi$

---

**Algorithm 5** train_task_policy()

    **Input** $p(\mathcal{M})$, task distribution; $\pi_\theta$, task policy; $\pi_\phi^{\text{explore}}$, exploration policy; $g_\omega$, encoder;
        $\hat{\boldsymbol{z}}_c$, shared component
    **Output** $\theta$, updated parameters

1:  $\mathcal{D} \leftarrow []$
2:  **for** $\mathcal{M}_i \sim p(\mathcal{M})$ **do**
3:     Sample exploration meta-episode $\mathbf{B}_j \sim P_{\mathcal{M}_i}^{\pi_\phi^{\text{explore}}}$
4:     $\boldsymbol{c} \leftarrow g_\omega(\mathbf{B}_j)$, using $\hat{\boldsymbol{z}}$ as the shared component
5:     $\mathcal{D} \leftarrow [\mathcal{D}, \tau \sim P_{\mathcal{M}_i}^{\pi_\theta(a|s,\boldsymbol{c})}]$
6:  **end for**
7:  **for** $j = 1, 2, \ldots, N_{\text{PPO}}$ **do**
8:     **for** $\tau \in \mathcal{D}$ **do**
9:        **for** $t = 1, 2, \ldots, H$ **do**
10:          Compute $V(s_t, \boldsymbol{c})$, $V(s_t)$, and $S[\pi_\theta](s_t, \boldsymbol{c})$
11:          $w(s_t, \boldsymbol{c}) \leftarrow$ compute using Eq. 3
12:          $r_t^+ \leftarrow r_t + \beta w(s_t, \boldsymbol{c}) S[\pi_\theta](s_t, \boldsymbol{c})$
13:          Compute and store $\hat{A}_t^{\text{GAE}}(\theta)$ using $r_t^+$
14:        **end for**
15:     **end for**
16:     Optimize $\theta$ on transitions from $\mathcal{D}$ using PPO
17: **end for**
18: **return** $\theta$

---

### D.1 Encoder

Both $f_u$ and $f_b$ use the same transformer architecture (Vaswani et al., 2017), except that $f_u$ is unidirectional, while $f_b$ is bidirectional. Each transformer has size $d_{\text{model}} = 128$, 8 layers, 16 attention heads, and feed-forward networks of size 512, with `GELU` activation functions (Hendrycks & Gimpel, 2016). All input timesteps are linearly mapped to $d_{\text{model}}$-dimensional embeddings, with positional encodings added before they are passed through the transformer. For stable training, we apply the T-Fixup initialization scheme (Huang et al., 2020), as recommended by Melo (2022).

The output of the transformer in $f_u$ is passed through a single-layer feed-forward network that outputs $\boldsymbol{\Gamma}$. Similarly, the transformer output of $f_b$ is independently processed by two separate single-layer feed-forward networks, outputting $\boldsymbol{z}$ and $\mathbf{Z}$. Each of these three networks has size 64 and uses `tanh` activation functions. When using data from multiple tasks, $\boldsymbol{\Gamma}$ and $\mathbf{Z}$ are computed separately for each task. In contrast, $\boldsymbol{z}$ is computed with data from multiple tasks, leading to a general representation of the entire distribution. Before computing $\boldsymbol{z}$, we apply average pooling to reduce the dimensionality of the output of $f_b$. We treat $\mathbf{Z}$ as a batch of $Q$ meta-episode representations, such that each slice $\mathbf{Z}_j$, for $j \in [Q]$, is computed independently by the feed-forward network. In a similar manner, each of the $Q$ meta-episodes has its own representation $\boldsymbol{\Gamma}$, computed independently by the feed-forward network in $f_u$.

The encoder $g_\omega$ has two final transformer layers, $h$, similar to those in $f_b$. The first layer combines $\mathbf{Z}$ and $\boldsymbol{\Gamma}$. The second computes $\boldsymbol{c}$ by combining the output of the first with $\boldsymbol{z}$. The reconstruction head $h_{\text{rec}}$ performs a linear transformation from $d_{\text{model}}$ to $d$. The task-reconstruction head $h_{\text{t-rec}}$ is a $(256, 128, 128)$ feed-forward network with `GELU` activations. We use the Adam optimizer (Kingma & Ba, 2015) to train $g_\omega$, $h_{\text{rec}}$, and $h_{\text{t-rec}}$.

### D.2 Policies

We represent $\pi_\phi^{\text{explore}}$, $V_\phi$, $\pi_\theta$, and $V_\theta$ using feed-forward networks. For task exploration, we use two separate $(128, 128, 128)$ networks with `tanh` activations to represent policy $\pi_\phi^{\text{explore}}$ and value function $V_\phi$. These take 64-dimensional embeddings of $s_t$ and $\boldsymbol{\Gamma}^{(:k,:t)}$ as input, computed by a single-layer, `tanh`-activated network. We use similar architectures for $\pi_\theta$ and $V_\theta$. However, $\boldsymbol{c}$ is first mapped to a 128-dimensional representation by a $(128, 128)$ feed-forward network with `tanh` activations.

To optimize both policies, each PPO update is run for $N_{\text{PPO}} = 4$ epochs, with minibatches of size 2. We use the Adam optimizer and set the coefficients in Eq. 12 to $c_1 = 0.5$ and $c_2 = 0.01$. The GAE $\hat{A}_t^{\text{GAE}}$ is computed using discounts $\gamma = 0.99$ and $\lambda = 0.9$. Finally, as mentioned in Sec. 3.1, we stabilize the optimization of $\pi_\theta$ by adding the PFO term suggested by Moalla et al. (2024) to the PPO objective (Eq. 12). We set the corresponding coefficient to $c_{\text{PFO}} = 0.1$. The PFO term is omitted when optimizing $\pi_\phi^{\text{explore}}$. We found that including it made training more difficult, whereas $\pi_\phi^{\text{explore}}$ was already sufficiently stable without it.

### D.3 Meta-Training

LaSER agents are meta-trained in two phases. In the first phase, we alternate between updating the encoder $g_\omega$ and the exploration policy $\pi_\phi^{\text{explore}}$ for $N_{\text{explore}}$ iterations. Before each encoder update, we collect a dataset $\mathcal{B}$ by following $\pi_\phi^{\text{explore}}$. Each element $\mathbf{B} \in \mathcal{B}$ is a tensor that contains $Q$ meta-episodes and is collected from a task $\mathcal{M}_i \sim p(\mathcal{M})$. A dataset $\mathcal{B}$ contains data from 5 different tasks. We update $g_\omega$ on the masked dataset $\mathcal{B}^{\text{masked}}$ for $N_{\text{encoder}}$ epochs, as shown in Alg. 3. To improve meta-training sample efficiency, we use all $Q$ meta-episodes in each tensor $\mathbf{B} \in \mathcal{B}$ to compute $\mathcal{L}_{\text{rec}}$ and $\mathcal{R}$. Additionally, in practice, each $\mathcal{B}$ is stored in a buffer and reused in future updates. This buffer holds data collected from up to $100,000$ tasks. To update $\pi_\phi^{\text{explore}}$, exploration data is collected from multiple tasks as described in Sec. 3.2. In the second phase, the task policy $\pi_\theta$ is updated for $N_{\text{task}}$ iterations.

For the loss $\mathcal{L}_{\text{rec}}$ used in Eq. 11, we exclude the reconstructed actions from the loss computation. This ensures that the encoder focuses on learning patterns in states and rewards, rather than the exploration policy. Additionally, we scale the loss on states by a factor of 0.05.

Note that the encoder $g_\omega$ is optimized purely through self-supervised methods. This constraint can be lifted. The RL loss used to update the task policy $\pi_\theta$ can also be used to fine-tune $g_\omega$. This fine-tuning could adapt the general pretrained encoder to align more closely with the task-solving objective. However, similar to Zintgraf et al. (2021b), we observe no empirical benefits from fine-tuning, so we omit it in our experiments.

LaSER's encoder $g_\omega$ and exploration policy $\pi_\phi^{\text{explore}}$ are meta-trained using an Nvidia A100 80GB GPU. In the second phase, we switch to an Nvidia RTX 4070 Ti SUPER 16GB GPU to meta-train the task policy $\pi_\theta$. All environment data is collected using only CPU computations.

## E    Meta-Training and Meta-Testing Tasks

We continue the discussion from Sec. 5.1.1 and provide more details on the tasks available in the MEWA benchmark. In a critical state $s_x$ of type $x$, a risky action $a_{\text{risk}}$ can either lead to a state $s'$ or $s'_x$. We use $s'$ to denote that a mistake has been avoided and the agent progressed in the task. In contrast, $s'_x$ denotes that a mistake has happened, resulting in a large delay in task completion and thus, lower returns. The probability of $a_{\text{risk}}$ leading to a mistake depends on both the mistake's type $x$ and the dynamics of task $\mathcal{M}_i \sim p(\mathcal{M})$. Formally, we define $T_i(s'_x \mid s_x, a_{\text{risk}}, \mathcal{M}_i) = \boldsymbol{p}_x^{(i)}$ to be the task-specific transition leading to a mistake. Here, the vector $\boldsymbol{p}^{(i)}$, with $\boldsymbol{p}_y^{(i)} \leq \boldsymbol{p}_x^{(i)}$ for all $y < x$, denotes the probabilities of making a mistake of each type. The probability of avoiding a mistake is then simply $T_i(s' \mid s_x, a_{\text{risk}}, \mathcal{M}_i) = 1 - \boldsymbol{p}_x^{(i)}$.

An agent's second option is to take a safe action $a_{\text{safe}}$ in a critical state $s_x$. This comes at the cost of a small delay, and leads to a critical state $s_{x-1}$, with $\boldsymbol{p}_{x-1}^{(i)} \leq \boldsymbol{p}_x^{(i)}$. Formally, the dynamics are given by the task-agnostic transition $T_i(s_{x-1} \mid s_x, a_{\text{safe}}, \mathcal{M}_i) = 1$.

Table 3 provides the 12 meta-testing tasks used for the results in Sec. 5. We describe each task by their corresponding vector of mistake probabilities $\boldsymbol{p}^{(i)} \in [0, 1]^4$. Additionally, for each task $\mathcal{M}_i$, we provide the average probability of a mistake of any type happening, computed as $1/4 \sum_{x=1}^4 \boldsymbol{p}_x^{(i)}$. As previously mentioned, we use this as a rough measure of similarity between tasks. Finally, we mark tasks that are outside of MEWA's meta-training task distribution (i.e., cannot be sampled during meta-training) as OOD tasks.

| Task | Mistake Probability | | | | Average Mistake | OOD |
| Index | Type I | Type II | Type III | Type IV | Probability | |
|---|---|---|---|---|---|---|
| 0 | 0 | 0 | 0 | 0 | 0.000 | No |
| 1 | 0.05 | 0 | 0 | 0 | 0.013 | No |
| 2 | 0.1 | 0 | 0 | 0 | 0.025 | No |
| 3 | 0.772 | 0 | 0 | 0 | 0.193 | No |
| 4 | 0.672 | 0.618 | 0 | 0 | 0.322 | No |
| 5 | 0.622 | 0.618 | 0.577 | 0.434 | 0.563 | No |
| 6 | 0.622 | 0.618 | 0.577 | 0.534 | 0.588 | Yes |
| 7 | 0.872 | 0.818 | 0.777 | 0 | 0.617 | Yes |
| 8 | 0.672 | 0.668 | 0.627 | 0.584 | 0.638 | Yes |
| 9 | 0.972 | 0.968 | 0.927 | 0.384 | 0.813 | Yes |
| 10 | 0.972 | 0.968 | 0.927 | 0.784 | 0.913 | Yes |
| 11 | 0.972 | 0.968 | 0.927 | 0.884 | 0.938 | Yes |

Table 3: The configuration of the 12 MEWA tasks used for meta-testing agents. A task is described by a 4-dimensional vector. Each dimension corresponds to the probability of a mistake of the corresponding type happening. We additionally compute the average mistake probability across all types. This can be seen as a way of comparing tasks, i.e., tasks with similar average probabilities have similar optimal policies. Finally, tasks marked as OOD are outside MEWA's meta-training task distribution.

## F   Baseline Algorithms Details

We provide details for the architectures, hyperparameters, and the meta-training process for the baseline algorithms used in Sec. 5.2 and 5.3. We tune these algorithms and use the best-performing hyperparameters. Where possible, and if performance is not negatively affected, the task policy's architecture and meta-training are similar to LaSER's. Otherwise, the task policy is tuned with the rest of the model. All baseline algorithms are meta-trained using an Nvidia RTX 4070 Ti SUPER 16GB GPU.

### F.1   MAML

For MAML (Finn et al., 2017), we use the publicly available code provided for meta-RL by Deleu (2018) at https://github.com/tristandeleu/pytorch-maml-rl. We train for 12000 meta-iterations, with batches of 16 tasks. For each task, we sample 5 episodes. MAML is meta-trained to maximize returns after one policy gradient update. However, during meta-testing, it performs $K = 4$ gradient updates. We use a discount factor of $\gamma = 0.99$. We also use $\lambda = 0.9$ to compute the generalized advantage estimator (GAE). To ensure optimal results, we did not use the first-order approximation proposed by Finn et al. (2017), but instead computed the second derivatives and backpropagated. The policy is a $(64, 64)$ feed-forward network with `tanh` activations. Due to the computational cost of MAML, we had to use a smaller policy network than in the other algorithms. All other hyperparameters follow the ones used by Finn et al. (2017) in their RL experiments.

### F.2   PEARL

We use the publicly available code at https://github.com/katerakelly/oyster for our implementation of PEARL (Rakelly et al., 2019). Each agent is meta-trained for 350 iterations on a total of 3000 training tasks, with 16 tasks per batch. At each iteration, both the encoder and the task policy are optimized for 2000 gradient steps, with batches of 64 transitions for the encoder and 256 for the policy.

PEARL uses a variational approach for computing task contexts $c$. When computing PEARL's loss, the KL divergence of the encoder is weighted by 0.1. Additionally, both the task policy and the encoder have a learning rate of $3 \times 10^{-4}$ and use the Adam optimizer.

PEARL uses Soft-Actor Critic (SAC) (Haarnoja et al., 2018a;b), an off-policy RL algorithm. Note that, since SAC is designed for continuous action spaces, we instead use a SAC version for discrete action spaces (Christodoulou, 2019). We tune SAC's temperature hyperparameter automatically, using the approach introduced by Haarnoja et al. (2018b), since manual tuning can be difficult. We use a discount factor of $\gamma = 0.99$ and a target smoothing coefficient for SAC of 0.005. At each iteration, PEARL adds data collected from 5 randomly selected tasks to a replay buffer of size 1000000 timesteps. For each of these tasks, it collects 400 timesteps by conditioning its policy on a context $c$ sampled from a prior distribution over tasks. It additionally collects 600 more timesteps using a $c$ sampled from its meta-trained posterior over tasks. However, this latter data is only used to update the task policy, not the encoder. Before training starts, the replay buffer is populated with 2000 timesteps per task, collected by following a uniformly random policy.

PEARL's encoder is a $(200, 200, 200)$ feed-forward network with `ReLU` activations that computes 5-dimensional latent task context vectors $c$. The task policy is a $(128, 128, 128)$ feed-forward network with `tanh` activations.

### F.3   VariBAD

We use the code at https://github.com/lmzintgraf/varibad for VariBAD (Zintgraf et al., 2021b). We meta-train for 8125 iterations on 3000 meta-training tasks. At each iteration, we collect 200 timesteps per task from 16 different tasks. These timesteps are stored in a buffer of size 10000 episodes. Additionally, before training, a uniformly random policy adds 5000 timesteps to the buffer. This buffer is later used to update the encoder.

VariBAD uses a variational auto-encoder (VAE) (Kingma & Welling, 2014) to encode the data collected from tasks into latent context $c$. This encoder is optimized using Adam with a learning rate of 0.001. At each iteration, the encoder is updated for 3 steps, using batches of 15 episodes, sampled from the buffer. The encoder is a recurrent neural network of size 128 that computes 5-dimensional task contexts $c$. The decoder takes $c$ as input, together with the current transition $s, a, s'$, and reconstructs the reward $r$. We use a $(64, 32)$ feed-forward network to represent the encoder. The state, action, and reward inputs are preprocessed into representations of size 32, 16, and 16, respectively, by separate single-layer networks with `ReLU` activations. Both the encoder and decoder have such pre-processing layers.

Since VariBAD uses PPO to optimize its policy, we use the same network architectures and hyperparameters as LaSER's task policy (see Appendix D.2 and Table 7). However, we do not use the additional PFO objective. Finally, before passing $s$ and $c$ to the policy, we embed them into 64-dimensional representations using separate single-layer networks with `tanh` activations.

### F.4 Decision Adapters

We implement the hypernetwork-based task policy using the code provided by Beukman et al. (2024) at https://github.com/Michael-Beukman/DecisionAdapter. The policy is a $(128, 128)$ feed-forward network, similar to LaSER's task policy (see Appendix D.2). It is also meta-trained with the same hyperparameters (see Table 7). However, this policy only takes the state $s$ as input. Moreover, between the policy's last hidden layer and output layer, we introduce an additional $(32, 32)$ network with `tanh` activations. The weights of this final network are generated by a hypernetwork, which takes the task context $c$ as input. Following Beukman et al. (2024), we also use a skip connection between the input and output of these hypernetwork-weighted layers.

The hypernetwork itself is a $(64, 64)$ feed-forward network with `tanh` activations. The output weights of the hypernetwork are computed in chunks of size 16. Before passing the input $c$, we pre-process it into a 4-dimensional representation using a single `tanh`-activated layer. Finally, the actor and the critic have separate hypernetworks.

## G Additional Results

We provide results that complement those in Sec. 5.

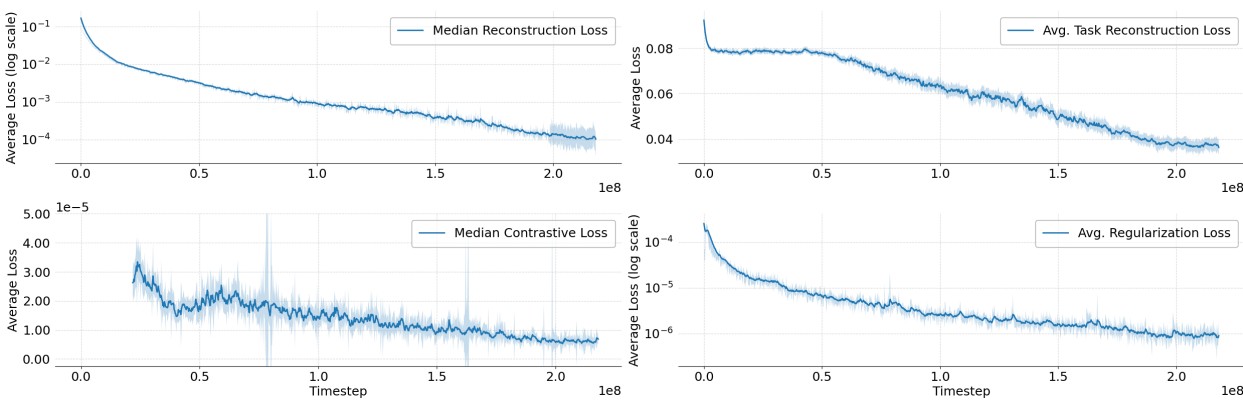

Figure 9: Encoder loss for LaSER, computed on tasks from MEWA's meta-training distribution. We show separate plots for each term in the encoder loss function. For 10 random seeds, we provide the standard error as the shaded areas, average the second and fourth metrics, and report the median for the other two. We found the median to be more appropriate, as one of the seeds was unstable towards the end of meta-training, for approximately $0.2e8$ timesteps. For this same reason, we restrict the plot of the contrastive loss to the range $[-2.5e{-}6, 5e{-}5]$. To aid readability for exponential losses, the reconstruction and regularization losses use a logarithmic scale. All metrics are smoothed using EMA with $\alpha_{\text{EMA}} = 0.2$.

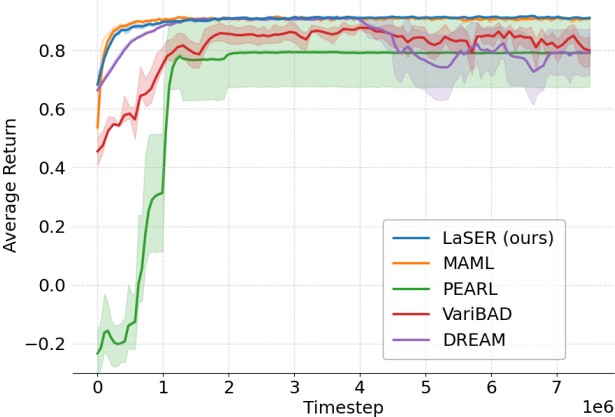

Figure 10: Undiscounted returns of task-solving policies on tasks from MEWA's meta-training task distribution. The shaded areas report the standard error of the average return across 10 seeds. We use EMA smoothing with $\alpha_{\text{EMA}} = 0.6$.

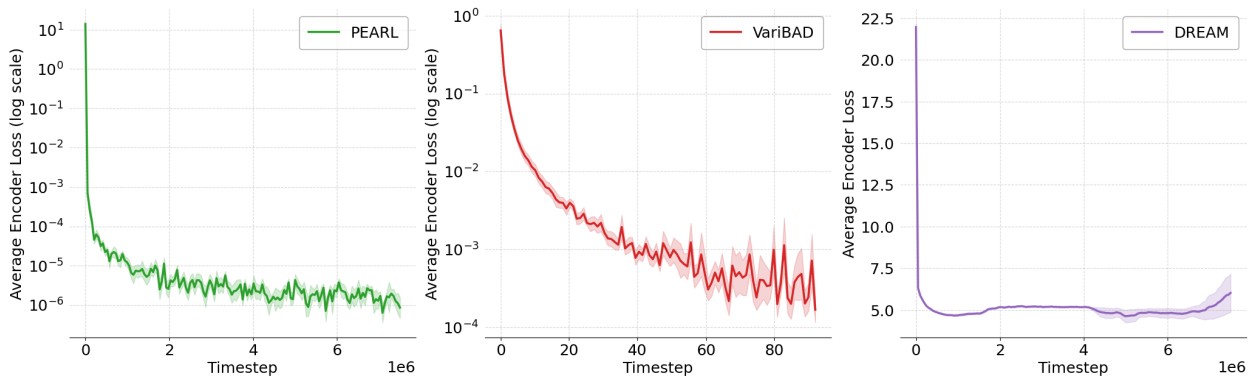

Figure 11: Encoder loss for PEARL, VariBAD, and DREAM on tasks from MEWA's meta-training task distribution. The shaded areas report the standard error of the average loss across 10 seeds. Since PEARL's and VairBAD's losses decay exponentially, we use logarithmic scales.

# H  Hyperparameters

| General Hyperparameters | | |
|---|---|---|
| **Hyperparameter** | **Value** | **Notes** |
| $N_{\text{explore}}$ | $10,000$ | |
| $H$ | $50$ | Horizon. |
| $K$ | $4$ | Episodes per meta-episode. |

Table 4: LaSER hyperparameters used throughout meta-training and meta-testing.

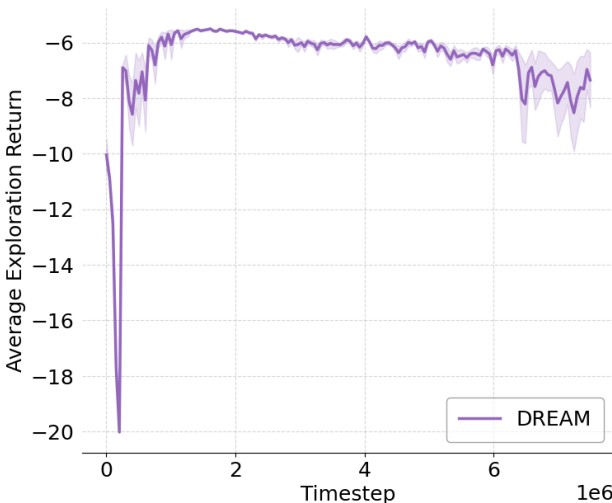

Figure 12: Exploration return for DREAM's exploration policy. Shaded areas are the standard error of the average over 10 random seeds.

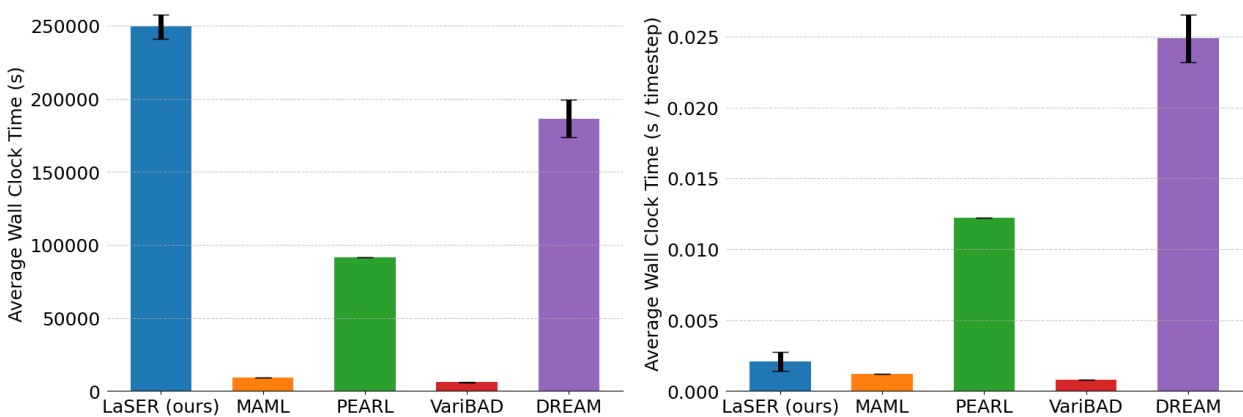

Figure 13: Wall clock time required to meta-train each method. We report standard error of the mean over 10 seeds. (Left) Total meta-training time. (Right) Average meta-training time per timestep.

| Encoder $g_\omega$ | | |
|---|---|---|
| **Hyperparameter** | **Value** | **Notes** |
| Learning rate | $1 \times 10^{-4}$ | |
| $N_{\text{encoder}}$ | 50 | |
| $Q$ | 20 | Meta-episodes per task. |
| $c_{\text{rec}}, c_{\text{t-rec}}, c_{\text{contr}}, c_{\mathcal{R}}$ | 0.5; 1; 1; 0.125 | Coefficients for loss $\mathcal{L}_{\text{LaSER}}$ in Eq. 11. |
| $\xi$ | 0.1 | Constant in Eq. 8. |
| $\nu$ | 250 | Update delay for parameters $\hat{\omega}$. |

Table 5: LaSER hyperparameters used for meta-training the encoder.

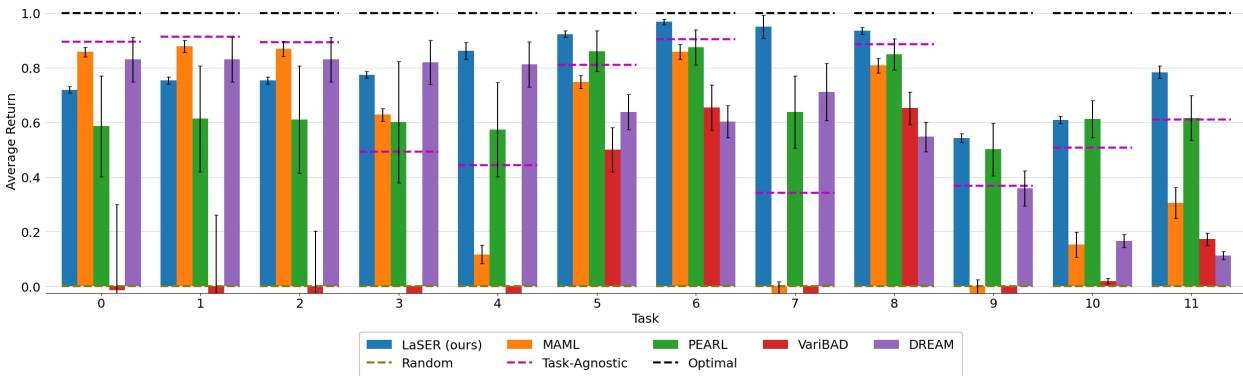

Figure 14: Average per-task returns for each of the 12 meta-testing tasks in MEWA. Each agent is meta-tested with an exploration budget of $K = 4$. Results are averaged over 10 seeds, with error bars indicating standard error across seeds. The three baselines are computed on a per-task basis. We normalize returns between the *random* and *optimal* baselines. For better visualization, we also restrict the plot to the $[-0.02, 1.0]$ range.

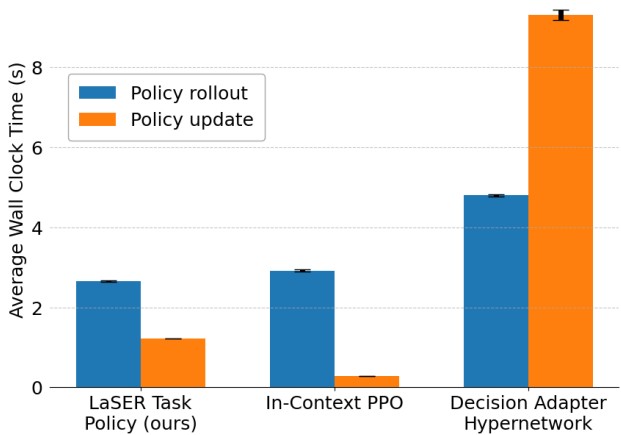

Figure 15: Wall clock time results for task policy meta-training. We measure the average time per iteration (in seconds) required to roll out episodes and optimize the policy. The results are averaged over 10 seeds. The bars represent standard error across all seeds.

| Exploration Policy $\pi_\phi^{\text{explore}}$ | | |
|---|---|---|
| **Hyperparameter** | **Value** | **Notes** |
| Learning rate | $1 \times 10^{-5}$ | |
| $\sigma$ | 0.0018 | Constant in Eq. 5. |
| $\epsilon$ | 0.1 | PPO clipping during exploration; see Eq. 13. |

Table 6: LaSER hyperparameters used for meta-training the exploration policy.

| Task Policy $\pi_\theta$ | | |
|---|---|---|
| **Hyperparameter** | **Value** | **Notes** |
| Learning rate | $1 \times 10^{-4}$ | |
| $N_{\text{task}}$ | $20,000$ | |
| $\beta$ | $1.5$ | Coefficient in Eq. 2. |
| $\zeta$ | $-0.13$ | Threshold in Eq. 3. |
| $\epsilon$ | $0.2$ | PPO clipping during task solving; see Eq. 13. |

Table 7: LaSER hyperparameters used for meta-training the task policy.

