# OpenReview forum: "Task-Specific Exploration in Meta-Reinforcement Learning via Task Reconstruction"
_TMLR — Rejected by TMLR_

### Review · Reviewer_Uy7r · 2025-03-22

**Summary Of Contributions:**

The paper proposes a new method for meta reinforcement learning. The proposed method, LaSER, includes: (1) A new reward function that helps the agent explore tasks by considering if the value function conditioned on the task representation is higher or not. (2) A new reward function specially designed to train the exploration policy. The reward function aims to make the agent explore episodes that differ from the explored ones using dot-product as the similarity. (3) A new method to train the task representation by finding a subset of episodes for a task can reconstruct most explorable episodes for the task by linear combination. The encoder model for encoding episodes into a task representation is chosen to be a combination of a unidirectional transformer and a bidirectional transformer.
Under the experimental setup of meta-training and meta-testing on the MEWA benchmark, the proposed method LaSER is shown to have better performance than MAML, PEARL, and VariBAD.

**Audience:**

Yes

**Broader Impact Concerns:**

I do not see a requirement for this work to include a Broader Impact Statement, as the paper is about a general algorithm without ethical implications.

**Claims And Evidence:**

No

**Requested Changes:**

* Clarity:
  * The presentation needs to be revised, at least without so much broken language, typos, and unpolished plots.
  * The raised questions about justifying method statements (listed in Weaknesses) need to be addressed in the paper to be clear and not misleading.
  * Move the Algorithm 1 and 2 to the main content.
* Experiments:
  * The experiments need to add baselines that use exploration policy, such as the ones that use contrastive learning to structure the exploration space, and more random seeds (currently is 5, which may not represent enough randomness).
  * Providing examples of the used environments in the main content.
  * Analyze and discuss the training time and testing time consumed by the proposed LaSER compared to the baselines.
  * Explain why the hypernetwork-generated policies are more unstable in Section 5.3 and why PPO+Hyper-Reward's learning curve is the slowest to rise.

**Strengths And Weaknesses:**

### Strengths

* The proposed method is interesting, has good intuition (although many assumptions may not be correct), and is perhaps useful. For example, the first part of the new reward function that considers the difference between $V(s,z)$ and $V(s)$, where $z$ denotes a task representation, perhaps can improve exploration in a simple way.
* The experiment setup does consider multiple details to justify the proposed method.

### Weaknesses
* The paper presentation needs to be polished a lot. There are many broken languages that prevent good understanding. For example,
  * The 2nd and 3rd sentences in Section 3.1 are not complete sentences on their own. A very similar pattern happens again at the 3rd paragraph in Section 3.1.
  * The 2nd and 3rd sentences in Section 5.3 are describing the same thing, likely to be accidental duplicates.
  * The 2nd paragraph of Section 5.3, “MEWA’a” => “MEWA’s”
* In addition, the paper’s writing, especially the method, is written in a flow that is hard to follow. The figures, plots, and algorithm blocks can also be rearranged. For example, why Figure 4 show learning curves of each method end at different timesteps? Why not show them all end at around $7\times 10^6$ timesteps? Also, Figure 6 can move the legends above both subfigures.
* Some method details justification:
  * In the 1st paragraph of Section 3, the authors mentioned “taking advantage of transformers’ strengths in in-context learning and long-sequence modeling.” However, much of the in-context learning ability is granted from pre-training instead of the architecture itself. The proposed method also does not seem to use in-context learning. Can the authors provide a reference to this statement and explain how their method utilizes in-context learning?
  * In the 2nd sentence after Equation 3, the authors mention, “increasing entropy is equivalent to increasing exploration.” However, while increasing entropy is often used to improve exploration, but they may not be equivalent, is there any proof or reference to justify this?
  * Why use dot product instead of cosine similarity for $d_k$ in Equation 4? The doc product also has the scale factor.
  * The authors mention “inspired by contrastive learning” above Equation 8, but the method seems quite different from contrastive learning. There is no pull away and pull closer mechanism for different tasks here. If there is a specific type of contrastive learning the authors mention here, it would be great to put the reference here.
* The experiment is not complete in demonstrating the method’s effectiveness. While the proposed method is mainly about exploration and representation reconstruction in meta RL, the compared baselines do not use exploration or representation. There should be other meta-RL methods, such as the ones the authors mentioned in Related Work.

---

> ### Author Response · Authors · 2025-06-13
>
> We thank you for your feedback and ideas. We believe we successfully addressed your concerns.
>
> Please see our general comment for a discussion of changes with respect to improvements on structure and the inclusion of additional baselines with a focus on exploration. We will address some of your more specific suggestions here.
>
> We rearranged some of the figures and algorithm blocks. Specifically, we moved algorithms 1 and 2 to the main text, we have moved the legend in Figure 6 (now 7). We have moved Figure 4 (now 5) to Sec. 5.1. Regarding this figure, we have also moved the convergence results of the baseline algorithms to the Appendix (Figure 10), and retrained all algorithms with the same amount of timesteps.
>
> We have also, in our opinion, improved the flow, making the text easier to follow, especially in the Methods and Experiments sections.
>
> We have discussed how LaSER is an in-context algorithm in more detail, by providing a more explicit discussion of in-context learning in the Preliminaries and Related Work sections. Specifically, we tied LaSER to the idea of in-context meta-RL [1], framing the exploration and task policies as in-context policies. From a different perspective, we have also explained how our encoders use in-context learning, which comes as a consequence of the transformer architecture we use. We explain the overlapping terminology between in-context meta-RL and in-context transformers in Sec 3.4, footnote 3.
>
> We have corrected and clarified our explanation of how entropy is used for meta-exploration, and added an explicit reference to the work of Williams & Peng, 1991 [2].
>
> Thank you for the suggestion of using cosine similarity instead of the dot product for learning task exploration in Equation 4 (now 5). We have followed this suggestion and re-ran all the experiments using cosine similarity, as well as adjusted the text in Sec 3.2 and throughout the paper to reflect this change.
>
> Eq. 8 introduced a contrastive learning approach to optimizing a latent space for task exploration. However, we agree that this was not properly discussed and was not obvious. We have therefore provided an explanation for how Eq. 8 relates to contrastive learning with a soft similarity signal. We have also discussed why this approach makes sense for translating our task reconstruction idea into a learning objective. These are in the last paragraph of Sec. 3.3.
>
> We have improved our description of the MEWA benchmark in the main text (see the new Sec. 5.1.1), providing better examples of what the tasks are, and how we perform meta-training and meta-testing.
>
> Finally, we have improved some of the explanations in the Experiments section. We provide additional detail on the learning of the hypernetwork-generated policies and discuss their instability, with references to the hypernetworks literature. We also provide a better explanation for why our task policy meta-training method is slower to rise, tying it to the extended meta-exploration the task policy performs. This is all discussed in Sec. 5.3.
>
> We have also increased the number of random seeds used in our experiments and reported in the results to 10.
>
> We hope our changes are satisfactory. We are looking forward to your response!
>
>
> References:
>
> [1] Moeini, A., Wang, J., Beck, J., Blaser, E., Whiteson, S., Chandra, R. and Zhang, S., 2025. A Survey of In-Context Reinforcement Learning. arXiv preprint arXiv:2502.07978.
>
> [2] Williams, R.J. and Peng, J., 1991. Function optimization using connectionist reinforcement learning algorithms. Connection Science, 3(3), pp.241-268.

---

### Review · Reviewer_nPi1 · 2025-04-02

**Summary Of Contributions:**

The authors propose a novel meta-reinforcement learning algorithm that focuses on two fundamental problems: learning exploration strategies for specific tasks and learning to effectively utilize task representations. First, the authors formalize two important assumptions about the data required for few-shot adaptation. Second, they leverage these representations to design an intrinsic reward for training a specific task. Meanwhile, they add a term to the standard reinforcement learning objective to encourage the efficient use of task descriptors for solving new few-shot tasks and introduce a hybrid encoder architecture. Finally, the authors demonstrate the effectiveness of their approach through experiments.

**Audience:**

Yes

**Broader Impact Concerns:**

No major concerns.

**Claims And Evidence:**

Yes

**Requested Changes:**

- I recommend the authors adjust the structure of the paper to make it easier to understand.
- In the related work section, it would be beneficial if the authors provide a comparative analysis with other meta-reinforcement learning algorithms that share similarities with LaSER in terms of exploration strategies and task representation learning. This would further highlight LaSER's innovative aspects and advantages.
- Add computational cost analysis.
- The author could consider conducting experimental validation of LaSER on more different types of task distributions to demonstrate the algorithm's universality and generalization ability.
- The authors may need to perform ablation studies to analyze the contribution of each key component (such as the task reconstruction module, mixed encoder, and hyper-reward mechanism) in LaSER to the final performance.

**Strengths And Weaknesses:**

**Strengths:**
- The LaSER learns how to identify and collect small but informative datasets from tasks through task reconstruction, and proposes a meta-learning hyper-reward mechanism. This idea and approach provide a new perspective for the field of meta-reinforcement learning.
- The author conducted detailed experimental evaluations of LaSER from multiple angles on the MEWA benchmark. The experimental results show that LaSER outperforms existing meta-reinforcement learning methods in various aspects, demonstrating the effectiveness of the algorithm.

**Weaknesses:**
- This paper has poor readability and lacks organization. Background knowledge and related work are interwoven with method design and experiments, disrupting coherence and logical flow. The paper's structure needs further improvement.
- The computational complexity is relatively high. LaSER involves multiple complex components and optimization steps during training, such as training the mixed encoder and calculating the task reconstruction loss. This may lead to high computational complexity, longer training times, and could limit its application to large-scale task distributions.
- Although the author conducts a thorough analysis in the MEWA environment, this work selects only one benchmark environment for algorithm performance testing. The algorithm's performance on other types of tasks has not been validated, and its universality and generalization ability require further investigation.

---

> ### Author Response · Authors · 2025-06-13
>
> We thank you for your feedback and ideas. We believe we successfully addressed your concerns.
>
> Please see our general comment for a discussion of changes with respect to improvements on structure, evaluating additional baselines and benchmark tasks, and runtime analyses. We will address some of your more specific suggestions here.
>
> We agree that our Methods/Experiments and Background/Related Work did have some overlapping framing, which can cause confusion. We have tried to improve the structure and in particular move more of the prior work discussion to the related work section. Specifically, we have moved the discussion on transformers from Sec. 3.4 to Related Work. We have also moved the discussion on in-context policies from Sec. 3.1 partly to Preliminaries and partly to Related Work. In the Methods and Experiments sections, we have only kept references that are directly relevant to our ideas, design choices, and evaluations.
>
> We have improved our Related Work section and discussed how LaSER differs from other methods in the literature in each of the three meta-RL phases: task-solving, exploration, and learning.
>
> With respect to the suggestion for additional ablation studies, we have tried to improve the argumentation in the paper that the experiments discussed in sections 5.3 and 5.4 have been performed to assess the specific impact of the different components of LaSER (task policy in Sec. 5.3 and exploration policy + task context encoder in Sec. 5.4). Since removing one of the proposed components will defy the applicability of the proposed architecture, we decided to perform test were we replaced them by ground truth information or adapted the task to focus on a specific subset. To our understanding, the insights generated through this can be considered at least equivalent to the contribution of "classical" ablation tests.
>
> We hope our modifications have addressed most or all of your main concerns. We are looking forward to your response!

---

> > ### Comment · Reviewer_nPi1 · 2025-06-18
> >
> > Thank the authors for their response. After reviewing their clarifications and the revised manuscript, most of my concerns have been addressed, but two issues remain:
> > - Computational Complexity: The authors did not address my question regarding computational complexity analysis. Clarification on this aspect would be valuable.
> > - Meta-World Benchmark: Before making my final recommendation, I would like to see experimental results on this benchmark to further validate the method's effectiveness.

---

### Review · Reviewer_smk9 · 2025-06-03

**Summary Of Contributions:**

The manuscript introduced a novel context-based meta-RL approach (called LaSER). Exploiting the transformer architecture and a few key assumptions, the approach learns a task representation which is used to condition the meta-RL task solving policy to facilitate adaptation. In addition, the approach induces exploration in the agent by exploiting and optimizing for dissimilarity between collected trajectories. The results showed that the approach achieved higher returns than the baselines it was compared against in the single benchmark (called MEWA) that was used.

**Audience:**

Yes

**Broader Impact Concerns:**

A broader impact statement of the proposed method should be discussed (for example, the use and effect of the method in robotic applications in real-world domains such as manufacturing).

**Claims And Evidence:**

Yes

**Requested Changes:**

1. Section 3.4, second paragraph: "While $Z_s$ and $\Gamma$ have similar roles, $\Gamma$ might fail to capture structure that is irrelevant for task exploration…". I think this should be “While $Z_s$ and $\Gamma$ have similar roles, $Z_s$ might fail to capture ….”
2. Section 5.3, first paragraph: "We meta-test task policies by rolling them out both before and after they receive a task descriptor z. During meta-testing, we roll out task policies both before and after receiving a task descriptor z.". This looks repetitive, consider deleting one of the sentences.
3. In the plots Figures 4 and 6, "The shaded areas represent the maximum and minimum across seeds". What does this mean? Is it the standard deviation or the confidence interval?
4. The link (https://anonymous.4open.science/r/LaSER-anon-4BBB) to the anonymized source code for the paper did not work when I tried opening it. I believe this can be easily fixed.

**Strengths And Weaknesses:**

**Strength**
1. A novel approach
2. Approach was well formulated and clearly described
3. The experiments support its claims.
4. A clear description of the hyper-parameters and implementation details of each method.


**Weakness**
1. Limited number of benchmarks. Only the MEWA benchmark was employed. The paper could be further strengthened by evaluating the proposed approach and baselines methods in other meta-RL benchmarks (e.g., meta-world)
2. Limited baselines: MAML, PEARL, and VariBAD was compared against the proposed approach. However, it would be useful to also compare against other approaches that address the issue of task exploration in meta-RL (i.e., the theme of the paper). For example, MAESN [1] and DREAM [5] could be good baselines since they focus on exploration. Also, since the proposed approach exploits the transformer network, it would be useful to compare against other transformer/attention-based meta-RL approaches (e.g., TrMRL [2] and SNAIL [3]). Lastly, why was the hypernetwork approach [4] not included as a baseline in Figures 4 and 5?
3. Computational complexity: Given the multiple neural components and step-wise training of the proposed approach, it would be useful to discuss about the computational complexity of the approach. What is the ratio of performance gain (over the baselines) to the computational complexity? Although an brief discussion was provided about the computational efficiency of the proposed approach in comparison to the task-conditioned hypernetwork approach [4], the comparison was done in a setup where only a subset of the neural components (i.e., only the task-learning policy) of the proposed approach was used. A report about the computational complexity that includes all neural components would be useful.


**References**
1. Gupta, A., Mendonca, R., Liu, Y., Abbeel, P. and Levine, S., 2018. Meta-reinforcement learning of structured exploration strategies. Advances in neural information processing systems, 31.
2. Melo, L.C., 2022, June. Transformers are meta-reinforcement learners. In international conference on machine learning (pp. 15340-15359). PMLR.
3. Mishra, N., Rohaninejad, M., Chen, X. and Abbeel, P., 2017. A simple neural attentive meta-learner. arXiv preprint arXiv:1707.03141.
4. Beukman, M., Jarvis, D., Klein, R., James, S. and Rosman, B., 2023. Dynamics generalisation in reinforcement learning via adaptive context-aware policies. Advances in Neural Information Processing Systems, 36, pp.40167-40203.
5. Liu, E.Z., Raghunathan, A., Liang, P. and Finn, C., 2021, July. Decoupling exploration and exploitation for meta-reinforcement learning without sacrifices. In International conference on machine learning (pp. 6925-6935). PMLR.

---

> ### Author Response · Authors · 2025-06-13
>
> We thank you for your feedback and ideas. We believe we successfully addressed your concerns.
>
> Please see our general comment for a discussion of changes with respect to improvements on evaluating additional baselines and benchmark tasks, and runtime analyses. We will address some of your more specific suggestions in the following.
>
> We respectfully believe that our original explanation of the differences between $Z_s$ and $\Gamma$ is correct, but perhaps unclear. We tried to convey that $\Gamma$ is optimized to learn representations that are only useful for task exploration. Therefore, it may miss certain patterns in the data if these are not useful during exploration. However, these might matter when computing task contexts, and $Z_s$ has a higher chance of recognizing these patterns. As a final note, to clarify notation, we have changed the name of $Z_s$ to $Z$.
>
> We have rerun our experiments and clarified that the shaded areas and error bars represent the standard error of the mean. We have also provided the std in the new Tables 1 and 2.
>
> Thank you for pointing out the broken link. It seems to have expired since the paper's original submission. The link in the paper is fixed now. We would also like to clarify that, once the paper is accepted, we will replace this with a link to our public GitHub repository.
>
> The Decision Adapters (DA) hypernetworks are only considered in Sec. 5.3 because they are designed to be only in-context policies that take ground-truth contexts as input [1], i.e., they are agnostic to how the contexts are actually produced or learned. Therefore, they are only concerned with the task-solving part of the meta-RL pipeline, so it does not make sense to compare them with the full LaSER algorithm. Moreover, the authors themselves report that DAs are sensitive to noisy (i.e., meta-learned) task contexts. However, we acknowledge that this might be confusing to the reader and we have therefore provided a footnote (6) explaining this.
>
> Your review also reports the need for a Broader Impact Statement. Could you please provide some additional context as to why you consider this necessary? Our understanding, based on the TMLR guidelines in Borader Impact Statements ("If their work carries a significant risk of harm, authors are required to include a Statement of Broader Impact" (https://jmlr.org/tmlr/author-guide.html)), was that this does not apply to work on general algorithms without real-world experiments on specific target applications. In particular, the overall impact of our method on such applications will be along the same lines as discussed for meta-RL per se, as the paper is mainly demonstrating superior performance in meta-RL scenarios. However, if you believe that there are specific broader impact effects or we should explicitly also report the impact of successful general meta-RL applications, we will try to provide some additional statements for the final version.
>
> We are hoping our improvements addressed your main concerns and we are looking forward to hearing back from you!
>
>
> References:
>
> [1] Beukman, M., Jarvis, D., Klein, R., James, S. and Rosman, B., 2023. Dynamics generalisation in reinforcement learning via adaptive context-aware policies. Advances in Neural Information Processing Systems, 36, pp.40167-40203.

---

### Author Response · Authors · 2025-06-13

We would like to thank each reviewer for their feedback. Your input has been valuable and we truly believe it has helped us improve our paper significantly.

We have uploaded a new version of the paper, where we have addressed your feedback.

Before addressing each individual review, we will first consider some of the common points.

To improve readability and general paper understandability, we have made significant changes to the paper's overall structure, as requested. For example, the introduction now discusses meta-RL along three main focus points. This should give a good intuitive starting point to readers unfamiliar with meta-RL. Additionally, this view allowed us to draw the reader's attention to which of the three components we discuss or evaluate.

The same aspects are also addressed in the restructured related work section, including a new section highlighting related work on in-context learning. In general, we tried to move more of the discussions referencing prior work to the related work section.

To improve the reader's intuition on how the meta-exploration bonus in Sec. 3.1 works, we have also added a short analysis of the gradient of the augmented RL objective.

We have addressed all minor corrections, like grammar, typos, repetitive phrases, and broken language. We have also changed the names of some of the variables we use, in the hopes of simplifying notation in both text and equations.

Regarding our results, we appreciate the suggestion of comparing LaSER with a meta-RL method that does task exploration differently. We have therefore added DREAM [1] as a baseline algorithm, which we believe to strengthen the results of our work. We discuss and evaluate DREAM in Secs. 5.1, 5.2, and 5.3. However, we would also like to point out that, while not designed solely for exploration, both PEARL [2] and VariBAD [3] introduced novel, explicit, approaches for performing task exploration in meta-RL, and are therefore good baselines for LaSER. They both also use representation learning for encoding contexts.

We have also added runtime evaluations for meta-training LaSER and compared this to each of the baseline algorithms (Sec 5.1.2).

We also understand why evaluating on a single benchmark can be considered a weakness. We had trouble finding good public benchmarks besides MEWA that provide explicit and transparent challenges for the meta-RL aspects that we wanted to address with our new algorithm. However, it is certainly important to also provide an understanding of the performance in some of the more general but widely used environments. We are therefore currently running additional experiments on Meta-World [4]. However, as this process is computationally expensive, we will need more time to include the results in the final paper update. We decided to provide an early update addressing all other issues, and hope the reviewers will accept to assess this point later on.

Additionally, we are planning to perform a small amount of minor changes in the Appendix, mainly focused on writing style. This is in addition to a missing appendix on meta-training details for Dream and a missing appendix for some simple derivations of the new gradient in Sec. 3.1. We hope the lack of these is not an inconvenience for the reviewers.

We will address the rest of the changes in the individual comments to each review. Thank you.


References:

[1] Liu, E.Z., Raghunathan, A., Liang, P. and Finn, C., 2021, July. Decoupling exploration and exploitation for meta-reinforcement learning without sacrifices. In International conference on machine learning (pp. 6925-6935). PMLR.

[2] Rakelly, K., Zhou, A., Finn, C., Levine, S. and Quillen, D., 2019, May. Efficient off-policy meta-reinforcement learning via probabilistic context variables. In International conference on machine learning (pp. 5331-5340). PMLR.

[3] Zintgraf, L., Schulze, S., Lu, C., Feng, L., Igl, M., Shiarlis, K., Gal, Y., Hofmann, K. and Whiteson, S., 2021. Varibad: Variational bayes-adaptive deep rl via meta-learning. Journal of Machine Learning Research, 22(289), pp.1-39.

[4] Yu, T., Quillen, D., He, Z., Julian, R., Hausman, K., Finn, C. and Levine, S., 2020, May. Meta-world: A benchmark and evaluation for multi-task and meta reinforcement learning. In Conference on robot learning (pp. 1094-1100). PMLR.

---

### Decision · Action_Editor_YwJA · 2025-07-25

**Recommendation:** Reject

**Additional Comments:**

The core idea of LaSER, combining transformer-based encoding with intrinsic rewards for task-specific exploration in meta-RL, is interesting and relevant to the TMLR community. However, the current submission lacks sufficient empirical validation and comparison with recent baselines, and omits key complexity analysis. Addressing these issues—particularly by completing experiments on additional benchmarks, including post-2021 baselines, and strengthening the evidence for claims—would significantly improve the paper. A thoroughly revised version could merit reconsideration.

**Audience:**

Yes

**Audience Explanation:**

The paper addresses core challenges in meta-reinforcement learning, such as task-specific exploration and context representation, which are of clear interest to the TMLR audience. The use of transformer-based architectures and the design of an intrinsic reward mechanism contribute to ongoing discussions in the field. Despite some limitations in empirical validation, the ideas and methodology are novel and relevant to researchers working on meta-RL, in-context learning, and representation learning.

**Claims And Evidence:**

No

**Claims Explanation:**

While the submission presents a novel meta-RL approach and provides some empirical support through results on the MEWA benchmark, the evidence is not yet comprehensive or fully convincing. Several reviewers noted critical limitations: the method is only evaluated on a single benchmark (MEWA), lacks comparisons to more recent baselines (post-2021), and does not include a sufficient analysis of computational complexity. Although the authors added one additional baseline (DREAM) and offered runtime comparisons, key concerns remained unresolved at the time of review. Therefore, the claims are only partially supported by clear and convincing evidence, but not to a level that satisfies the standard expected for publication in TMLR.

**Resubmission Of Major Revision:**

The authors may consider submitting a major revision at a later time.